# Primary school staff perspectives of school closures due to COVID-19, experiences of schools reopening and recommendations for the future: A qualitative survey in Wales

Emily Marchant[1]*, Charlotte Todd[1], Michaela James[1], Tom Crick[2], Russell Dwyer[3], Sinead Brophy[1]

1 Medical School, Faculty of Medicine, Health and Life Science, Swansea University, Swansea, United Kingdom, 2 Department of Education & Childhood Studies, Faculty of Humanities and Social Sciences, Swansea University, Swansea, United Kingdom, 3 St Thomas Community Primary School, Swansea, United Kingdom

* E.K.Marchant@swansea.ac.uk

## Abstract

School closures due to the COVID-19 global pandemic are likely to have a range of negative consequences spanning the domains of child development, education and health, in addition to the widening of inequalities and inequities. Research is required to improve understanding of the impact of school closures on the education, health and wellbeing of pupils and school staff, the challenges posed during face-to-face reopening and importantly to identify how the impacts of these challenges can be addressed going forward to inform emerging policy and practice. This qualitative study aimed to reflect on the perspectives and experiences of primary school staff (pupils aged 3–11) in Wales regarding school closures and the initial face-to-face reopening of schools and to identify recommendations for the future. A total of 208 school staff completed a national online survey through the HAPPEN primary school network, consisting of questions about school closures (March to June 2020), the phased face-to-face reopening of schools (June to July 2020) and a return to face-to-face education. Thematic analysis of survey responses highlighted that primary school staff perceive that gaps in learning, health and wellbeing have increased and inequalities have widened during school closures. Findings from this study identified five recommendations; (i) prioritise the health and wellbeing of pupils and staff; (ii) focus on enabling parental engagement and support; (iii) improve digital competence amongst pupils, teachers and parents; (iv) consider opportunities for smaller class sizes and additional staffing; and (v) improve the mechanism of communication between schools and families, and between government and schools.

## Introduction

The COVID-19 global pandemic caused by the severe acute respiratory syndrome coronavirus 2 (SARS-CoV-2) resulted in the temporary closure of education across all settings and a rapid

**Data Availability Statement:** The data from this research study are not publicly available due to concerns of participant confidentiality. Data from

this research study contain information that are identifiable at both the school and individual level. Ethical approval for this research study was granted by the College Medicine Research Ethics Committee, Swansea University (approval number 2017-0033D, email sumsresc@swansea.ac.uk), on the basis that participants' data was only accessible by the research team. The participants did not consent to having their data publicly available. Requests for access to data may be directed to the Swansea University Medical School Research Ethics Committee, to be considered on an individual basis, by emailing sumsresc@swansea.ac.uk.

**Funding:** This work was supported by the National Centre for Population Health and Wellbeing Research (https://ncphwr.org.uk/). The Economic and Social Research Council (ESRC) funded the development of the HAPPEN network (grant number: ES/J500197/1; EM was the recipient) which this research was conducted through.

**Competing interests:** The authors have declared that no competing interests exist.

move to "emergency remote teaching". School closures were implemented as a public health measure to reduce social contacts and transmission of SARS-CoV-2 in countries worldwide. At its peak, school closures were experienced by over 1.6 billion pupils from more than 190 countries globally [1].

Children under 14 years appear to have lower susceptibility of infection than adults and school-aged children display a reduced risk of clinical symptoms [2–6]. A rapid systematic review assessing the effectiveness of school closures during past and present coronavirus outbreaks suggested that school closures are likely to have only a relatively small impact on reducing overall transmission [7]. Among the 16 studies considered, the review concluded that the current evidence to support national school closures is weak given the characteristics of COVID-19 dynamics in school-aged children.

School closures have wider and unintended consequences spanning the domains of child development, health, social and economic outcomes [8–10]. Globally, it is estimated that school closures result in an average loss of between 0.3 and 0.9 quality-adjusted years of schooling, contributing to a reduction in potential earnings during adulthood [11]. Closing schools also cause economic harms at a family and societal level through a loss of parental productivity, earnings and childcare responsibilities [12]. Prolonged periods away from school demand the provision of adapted education practice, which requires significant parental input, engagement and a positive home learning environment [13]. In addition, negative impacts on learning progression, social interaction and physical and mental health have also been demonstrated [14].

Most notably, a consequence of school closures is the widening of inequalities in children's health and education outcomes. Survey research from the UK suggests that over half of teachers report that the learning gap between disadvantaged pupils and their peers has widened. Estimates suggest a 46% increase in this curriculum learning gap [15]. Widening educational inequalities have also been observed internationally including across Europe [16, 17] and the USA [18].

In weighing up the benefits and risks of school closures, experts concluded that keeping schools open should be a national priority, and given both the immediate and longer term negative consequences closing schools should be a last resort [14, 19]. There is currently little evidence regarding the challenges posed to schools during school closures and reopening face-to-face. There needs to be research examining the impact of reopening schools not just on the epidemic curve, but also on the education, health and wellbeing of pupils and school staff [20, 21]. It needs to acknowledge the challenges posed during reopening and how countries can return to a "*new (ab)normal*", informing emerging policy and practice [22, 23] Furthermore, it is important for rapid research conducted during the COVID-19 pandemic to be shared as "*actionable findings*", recognised by Vindrola-Padros and colleagues as "*straightforward recommendations that can be easily understood and translated into changes in policy and/or practice*" [24].

In Wales, one of the four constituent nations of the UK with a population of 3.17 million, education is a devolved responsibility of the Welsh Government. The announcement of school closures came into effect on 20 March 2020, alongside the rapid development and implementation of a "*continuity of learning*" policy [25]. This "*Stay Safe. Stay Learning*" strategy focused on the safety, physical and mental health and wellbeing of learners and the wider education workforce, as well as providing the principles for maintaining learning, and the transition back to schools at the appropriate time [25]. Following a lockdown period of over 14 weeks, schools reopened using a phased approach with limited capacity on 29 June 2020 for a period of "*check in, catch up and prepare*" until 27 July 2020. All schools in Wales reopened from 1 September 2020 for a two week period to return to full capacity, with statutory education resuming on 14

September 2020. The delivery of an adapted approach to teaching, learning, and school operation and the continual management of risk are required whilst also addressing the impact and challenges as a result of school closures. These include the use of contact group 'bubbles', limiting contact between groups, social distancing between staff, pupils, parents and visitors, and the requirement of group isolation following COVID-19 outbreaks supported by the "*Test*, *Trace*, *Protect*" strategy [26, 27]. Schools have also adopted a hybrid/blended learning approach through a combination of in-school and remote teaching when necessary.

The implementation of national school closures as a public health measure to reduce social contacts has resulted in a range of unintended consequences. Evidence demonstrates the impact on children's development, physical and mental health, social and economic outcomes and widening inequalities [8–12, 14, 15]. However, there is limited evidence regarding the challenges posed to schools during school closures and reopening, and a gap in understanding of the strategies for schools returning based on these impacts [22]. This study therefore aims to address this gap in evidence by exploring what were the challenges posed to schools during school closures and reopening and how can the impact of these challenges be addressed going forward.

Grounded theory is a useful approach when little is known about a phenomenon [28]. In this case of this study, the rapidly evolving impacts of the COVID-19 pandemic brought with it unprecedented changes to society including national school closures based on limited evidence of potential challenges facing schools and learners. This study uses an interpretive approach with tenets of grounded theory methodology in order to develop a pattern of meanings, explanations and descriptions that reflect the perspectives and experiences of primary school staff within the context of school closures and phased reopening of schools. The purpose of the theoretical approach used in this study is to explore and extract the different perspectives in order to develop a rigorous and robust foundation of "*actionable findings*" [24] for shaping and influencing emerging policy and practice in post-COVID education provision in Wales and more broadly across the UK. Thus, the pragmatic approach to grounded theory in assessing and addressing challenges in practice [29] is suitably aligned to the aims of this study.

The aim of this grounded theory qualitative study is therefore to reflect on the perspectives of primary school staff regarding school closures between March and June 2020 and their experiences of the initial phased face-to-face reopening of schools between June and July 2020 to identify recommendations for the face-to-face reopening of schools from September 2020 or future scenarios of school closures and blended learning.

## Methods

This study adopted a qualitative design in order to understand the perspectives of primary school staff (working with pupils aged 3–11) regarding school closures and the face-to-face reopening of schools within Wales, UK. The basis of this study relies on an interpretivist approach, whereby the authors assume social reality is shaped by human experiences (primary school staff) and social contexts (primary school closures). Grounded theory research gathers data relating to participants' different lived experiences in order to generate plausible theory to understand the contextual reality of common circumstances [30]. Tenets of a grounded theory approach were applied to the data collection, analysis and explanatory findings in order to offer descriptions of the perspectives and experiences of primary school staff during the period of national school closures (between March and June 2020) and the phased reopening of schools (June to July 2020).

Generating theory in grounded theory relates to plausible relationships proposed amongst concepts and sets of concepts [31]. In the case of the current study, the generation of theory

relates to the development of recommendations, viewed as "*actionable findings*" [24] regarding the face-to-face return from September 2020, based on these lived experiences of primary school staff. Therefore, this approach intends to inductively construct these recommendations aligned to these lived experiences to contribute to the limited evidence base of the impact of school closures and to shape and influence emerging policy and practice during the 'education recovery' period following the COVID-19 pandemic.

### School staff survey

An online survey with primary school staff was distributed in July 2020 through the HAPPEN primary school network (*Health and Attainment of Pupils in a Primary Education Network*) [32]. The survey was open for responses between 9–31 July 2020. Inclusion criteria for participating in the survey was any primary school staff member (headteachers, deputy headteachers, teachers, teaching assistants, support staff) working within a local authority maintained primary school in Wales, UK. A grounded theory strategy using thematic analysis was used to generate themes of recommendations based on survey responses to gain an understanding into the perspectives and experiences of primary school staff during the period of school closures (March to July 2020) and the phased reopening of schools (June to July 2020) to inform the face-to-face return to education from September 2020.

Rapid development of the survey was required due to limited notice on changes to education provision, including the announcement of the return to school. The development of the survey questions was based on input from the research team specialising in child health and education research (authors), key stakeholders in education (two regional education consortia curriculum staff) and two school staff from different primary schools (one teacher, one headteacher). Following revisions and refinement from key stakeholders and primary school staff, the survey was piloted with the school staff to ensure wording and usability. The final survey contained 25 questions in total, consisting of a combination of categorical questions relating to demographics (school name, role, year group), school reopening logistics (e.g. full/part-time structure, class sizes, overall attendance proportion, outdoor and play provision) and support/training requirements (e.g. professional development training and support). Given that interpretivist grounded theory is based on inductively developing patterns of meaning, a range of broad, open-ended questions were included to explore the research question. As grounded theory presumes limited preconceived ideas of the research topic, the survey incorporated polarised open-ended questions regarding school closures (e.g. benefits/negative impacts of school closures, concerns and recommendations for face-to-face reopening). These questions were open-ended in order to gather information rich narratives of participants' perspectives, experiences and recommendations [33].

A copy of the full survey is available in the S1 File. For the purpose of this research study, responses relating to lockdown/school closures were gathered from questions 18, 19 and 20, and responses relating to the face-to-face return to education were gathered from the questions 21, 22 and 23.

### Participants

A convenience sample of staff (headteachers, deputy headteachers, teachers, teaching assistants, support staff) working in primary schools across Wales were invited to participate in the HAPPEN School Staff survey in July 2020. A combination of recruitment methods were used in order to increase the reach and uptake of the survey aside from pre-existing schools engaged with HAPPEN. Initially, all primary schools in Wales (which include pupils from ages 3–11) were contacted via email through HAPPEN inviting their staff to complete the survey. A

snowball sampling method was used following initial contact in which the representative from each school was asked to share the survey with staff within the primary school. Next, the survey was shared with key education stakeholders (e.g. local authorities, regional education consortia) to disseminate through their networks and existing partnerships. Finally, a social media campaign was delivered using paid advertisement on Facebook and Twitter.

A total of 208 staff from 78 primary schools across 16 local authorities completed the survey (8% did not provide details on school name or local authority). Reasons for non-participation were not captured at a school or individual staff level. The sample consisted of 20 headteachers, 9 deputy headteachers, 106 teachers, 54 teaching assistants, 3 higher level teaching assistants, 9 support staff and 6 other (e.g. supply teacher, special needs/additional learning needs teacher). The survey was open to staff across all years of primary school (reception: ages 4–5; year 1: ages 5–6; year 2: ages 6–7; year 3: ages 7–8; year 4: ages 8–9; year 5: ages 9–10; year 6: ages 10–11). Of those that provided information on year group, 41% worked within the Foundation Phase (ages 3–7) and 59% in Key Stage 2 (ages 7–11). 84% of the sample provided school name. This was used to manually obtain information regarding the proportion of pupils eligible for free school meals (FSM) using the 'My Local School' government website [34], and ranged from 1.7–58.7% (national average 19%). Eligibility criteria for FSM in Wales is based upon households receiving state benefits such as Income Support and FSM is commonly used as proxy measure of socio-economic disadvantage [35, 36]. 50% of school staff worked in schools with a proportion of pupils eligible for FSM greater than the national average (>19%).

## Ethics

Information sheets and consent forms were distributed to participants detailing the aims of the study. To participate in the survey, primary school staff were required to provide written informed consent. Ethical approval was granted by the Swansea University Medical School Research Ethics Committee (approval number: 2017-0033D). All participants were able to withdraw from the research at any point. All personal data such as school names were anonymised; electronic data (survey responses) were stored in password protected files only accessible to the research team.

## Data analysis

This research study adopted an interpretivist approach through thematic analysis of open-ended survey responses. Thematic analysis is considered a suitable approach in gaining meaningful comprehension of patterns in participant perspectives captured using a range of qualitative methods including open-ended survey questions [37, 38]. Thus, this study used thematic analysis as a data analysis technique to elicit participants' perspectives and experiences relating to school closures and the initial phased reopening of schools. In providing context-specific explanations and descriptions of the impacts of school closures, this study used tenets of a grounded theory methodology to inductively develop theory presented as recommendations to inform the face-to-face return to education and emerging policy and practice. Thematic analysis as an analytical approach in grounded theory studies has been used widely in other settings-based research [29].

The lead (EM, PhD) and second (CT, MSc) researcher involved in data analysis were both female and had previous experience in conducting thematic analysis in qualitative school-based research. The researchers did not have any interaction with participants. Responses from the school staff survey were downloaded to an Excel spreadsheet. Participants were assigned a unique study ID and identifiable information removed. The coding process followed the six steps outlined by Braun and Clarke [38]. Firstly, the lead researcher (EM)

systematically read responses several times to facilitate immersion in the data. Re-reading responses allowed familiarity with the data, and the use of memoing recorded notes of patterns and emerging insights relating to coding ideas. Thoughts relating to decision processes were documented in a reflexive journal [39, 40]. These documented notes formed the basis of the second stage of coding, whereby initial codes using words and phrases were manually assigned to participants' responses. The second researcher (CT) independently coded random survey responses from 50 participants. These coding responses were compared between researchers to account for consistencies or inconsistencies in their interpretations of responses and associated codes. Agreement between researchers was very high; if there was a discrepancy or disagreement, a third researcher adjudicated (SB). This method attempts to reduce researcher bias and enhances the validity of codes and categories assigned [41]. Following this, the third stage of coding involved the two researchers working together through an extensive process of collating, combining and categorising codes into over-arching themes and sub-theme headings. This stage incorporated a grounded theory approach whereby both researchers revisited the coded survey responses and followed a constant comparative method to progressively categorise and refine the themes and sub-themes. In cases where codes were merged, removed or renamed, notes were taken in a reflexive journal and raw files were saved as new versions and archived with dates in order to keep an audit trail of the coding process [40]. Throughout these processes, a codebook and manual was developed to record codes and themes. The themes and sub-themes were further refined in stage four to ensure clear distinctions between themes. These themes were then grouped into over-arching recommendations for future school closures and the reopening of schools. The lead researcher then manually reviewed the open-ended survey responses and coded according to the final list of recommendation themes and sub-themes. These responses containing themes and sub-themes were compiled to a master copy document that was used as a reference for stages 5 and 6 whereby themes were written and discussed within the findings.

To ensure trustworthiness, the grouping of survey responses into recommendation themes developed by the researchers were screened by a primary school teacher. The interpretation and discussion of themes and the final manuscript was also peer-reviewed for member checking by a primary school headteacher (study participant and author, RD) for corrections, clarification or confirmation [40, 42].

## Results

### Prioritise the health and wellbeing of pupils and staff

A key priority highlighted by school staff was the importance of prioritising the health and wellbeing of pupils and staff. In summary, school staff consistently mentioned a range of negative impact of school closures on children. This included physical health concerns such as decreased physical activity and fitness, weight gain and lethargy, mental health concerns including increased anxiety and lower wellbeing, and the impact on social development such as a lack of social interaction, excessive gaming and social media use. Additional challenges were also highlighted regarding children with additional learning needs (ALN). The impact on school staff wellbeing included challenges with work/life balance such as the pressure of providing online teaching and learning whilst balancing personal issues and family life. Therefore, the recommendation to prioritise the health and wellbeing of pupils and staff included suggestions of specific wellbeing activities for pupils, ensuring there are designated staff responsible for wellbeing, protecting staff breaks and increased provision of outdoor learning and play. This was also emphasised in relation to staff perceptions that the return of face-to-face education would prioritise the rhetoric of "*catching up*" through intensive teaching and learning at

the detriment of whole-school health and wellbeing. This will be discussed in more detail below.

Whilst on one hand, positives to school closures were highlighted by some school staff relating to potential benefits to children's wellbeing through strengthened family relationships and activities; "*Spending time with family, baking, walking etc has been hugely beneficial to their wellbeing*" (headteacher), there were also significant concerns expressed regarding wellbeing, anxiety and general mental health problems. One headteacher incorporated quantitative measures of children's wellbeing during lockdown and reported that "*assessments undertaken since lockdown indicate a decline in emotional well-being*". In addition, of the categorical responses asking school staff what professional development or training staff required upon the return to school, 62% chose '*Supporting learning health and wellbeing*', highlighting the demand for additional support within the school setting.

Many teachers perceived that pupils were engaging in less physical activity during lockdown, with responses to the negative effects of school closures including "*less active*" (year 1 teacher), and a "*lack of physical fitness*" (reception teacher). In addition, weight gain was observed by a number of teachers upon the phased return to school; "*some children have put on an unhealthy amount of weight*" (year 2 teaching assistant). Staff also perceived increased screen time and gaming behaviours with responses including "*a lot of time spent on computers and not going outside*" (year 2 teaching assistant), and "*Playing inappropriate games and excessive screen time*" (year 3 teacher).

During school reopening, school staff observed that "*some seem lethargic coming into school and parents report poor sleep routines*" (headteacher). Others perceived that the lack of school routine impacted subsequent class behaviour, stating there were "*discipline issues with pupils not having the school structure for a long time*" (year 3 teacher). Staff working with children with ALN highlighted the challenge of a lack of school structure; "*ALN children especially finding this time difficult—Struggles due to lack of routine for many*" (year 2 teacher). Others outlined their expectations for the face-to-face return to school; "*the children will have been affected by being out of school for so long, and will have trouble re-adjusting to school life*" (year 2 teacher).

Staff were concerned about the lack of social interaction during the period of school closures, with those teaching younger year groups referencing pupils to be "*isolated*" at home (reception teacher), "*distanced from peers*" (year 2 teacher) and commenting that "*children reluctant to speak in group contexts*" (year 3 teacher). Some teachers had received concerns from parents regarding their children; "*Some parents have informed us that their children are quite withdrawn and have stopped connecting with friends*" (year 4 teacher), and other teachers highlighted similar experiences upon the return to school; "*Some pupils are quieter and withdrawn*" (year 6 teacher).

Therefore, a key recommendation for future scenarios of school closures or remote learning (e.g. self-isolating) was the importance of maintaining and promoting good health and wellbeing. This included suggestions of "*more wellbeing phone calls to children and parents*" (reception teacher). A number of teachers suggested that this should be prioritised immediately for example through "*wellbeing checks from the start, more interaction with children from the start*" (year 1 teacher).

Whilst staff highlighted concerns about learning regression, discussed in Recommendation 2: *Focus on enabling parental support and engagement*, prioritising health and wellbeing was still advocated for upon the return to school, particularly with the uncertainty of the longer-term impacts of COVID-19 on the wider education system and expectations by staff; "*there might be a push on rushing straight back into learning at a fast pace where I think the health and wellbeing needs to be the main focus*" (higher level teaching assistant). For example, a teacher

called for "*the ability to support pupils emotionally as well as forwarding their education*" (year 5 teacher). Suggestions of removing pressure from formal tests and assessments in order to focus on wellbeing; "*Support from senior management to allow us to nurture the children and settle them before they start learning. Less pressure about assessments and attainments and more focus on the children and their needs and feelings*" (higher level teaching assistant). Removing this pressure placed upon school assessments and attainment was highlighted by a range of school staff, with responses including "*less emphasis on school results/comparing of schools attainment/moderation*" (deputy headteacher), and "*no target setting or data collection next year*".

Recommendations for prioritising health and wellbeing within the school environment included "*planned activities around the well-being of children*" (year 3 teaching assistant) and "*designated staff in each school to help with well-being*" (year 3 teacher). During the phased return to school, some schools offered increased provision of outdoor learning and play in addressing wellbeing; "*Timetabled regular outdoor sessions, some planned, some free play*" (reception teacher). The requirement of additional support and infrastructure were suggested in increasing outdoor provision in the face-to-face return to education; "*more adults to help take groups outside to work on health and well-being*" (year 6 teacher) and "*we need outdoor shelters so we can do more outside work and the bubbles*" (year 2 teacher).

School staff also raised concerns that their own wellbeing had been ignored during the period of school closures and returning to school. School staff highlighted challenges they had experienced when combining work and home life; "*the lines blurred with work life balance*" (headteacher). Suggestions about future potential school closures included "*less pressure on teachers to continue to provide the same level of work whilst working from home trying to support our own families*" (year 4 teacher), "*flexible working for teachers*" (year 4 teacher), and ensuring that communication from schools between staff was limited to working hours.

Upon the phased reopening, school staff commented that "*all stakeholders don't seem to be considering the staff's well-being and worries and anxieties about the whole school return*" (year 6 teacher), and that there were "*staff anxieties with more children and less social distancing*" (headteacher). Concerns were also raised about staff wellbeing in relation to adhering to social distancing and the lack of contact with other school staff; "*Staff isolation—the very nature of primary school approach is collaboration and support. . .But this is why I mentioned 'staff isolation' and protected breaks—it's an issue*" (year 6 teacher). Therefore, within the school day, staff recommended "*protecting staff breaks (which is essential and hasn't happened during 'return to school')*" (year 6 teacher). Regarding the face-to-face return to education, a headteacher emphasised that "*Government need to address teacher anxiety. We need to know that our health is also a priority*".

Fears regarding transmission within the school setting upon the return to school were highlighted consistently. Concerns were also raised by those deemed as high risk, or with vulnerable family members; "*As a vulnerable member of staff I fear going back with a full class and spending the day trying to distance myself. How can I teach properly like that?*" (year 1 teacher) and another stating "*I risk taking the virus home to members of my family who are extremely vulnerable*" (support staff). Staff felt that scientific evidence would make the face-to-face return to school easier; "*Having more concrete medical information stating that I am not putting my life at jeopardy being amongst so many and not being able to socially distance*" (year 1 teaching assistant).

Recommendations to address these concerns included regular and increased availability of testing for pupils and school staff, with responses including "*routine testing of school staff including being given an anti-body test*" (year 3 teacher) and "*regular and continuous testing in schools, and strict self-isolating for bubbles*" (year 4 teacher). Furthermore, school staff

emphasised the importance of having effective test, trace and protect programmes; "*A more robust track and trace system where people are informed of numbers in areas etc*" (deputy head-teacher). There was indeed overlap here with recommendation 5 regarding clearer guidance and communication.

### Focus on enabling parental support and engagement

The importance of the home learning environment during school closures was highlighted as a key factor in children's learning progression. This is outlined in the findings from this recommendation theme in which primary school staff noted widening inequities and inequalities. Staff in this study commented that the large variation in home learning environments and parental support/engagement accounted for gaps in children's learning. This theme encompasses aspects of parental support and engagement that are both positive; one-to-one support, wider skill development, improved parent/school relationships, parental awareness of children's learning needs, and negative; key skill regression, lack of engagement with/decreased motivation for learning. School staff acknowledged a wide range of barriers for parents and families supporting home learning, including working commitments and language barriers. Therefore, staff suggested introducing support systems such as advice helplines for parents, and emphasised the importance of strengthening links between schools and families.

Benefits cited by school staff included children receiving one-to-one support from parents, and positive examples of parental engagement were discussed; "*parents more aware of the learning needs of their children*" (year 3 teacher) and a sense of "*parents' realisation of what pupils are taught*" (year 3 teacher). In the case of high levels of family support, school staff reported improvements to children's wider development, enhancing skills not covered in the traditional curriculum; "*Some are learning new skills not taught in primary schools*" (reception teaching assistant) and through practical opportunities within the home; "*Lots of children have learnt life skills such as cooking, washing and budgeting*" (year 2 teaching assistant) and outdoors; "*They learnt new life skills like riding bikes*" (year 3 teaching assistant).

Experiences of high parental engagement were also reflected in improved relationships with and communication between the school and parents; "*We've established an excellent working relationship with parents. Parents have taken more responsibility over their children's learning and wellbeing*" (headteacher) and school staff commented; "*A massive improvement witnessed in the communication between myself, school and parents*" (reception teacher). For some, this also enabled parents to communicate with schools regarding extra support for their children; "*Parents have become more aware of what their children can and cannot do and have sought advice about how to address the areas of concern*" (year 4 teacher).

However, school staff reported wide variations in home learning provision; "*For those children who have had parents who have supported them with online learning this has been great*". Discussions on this topic included the inequalities and inequities present and a sense of widening gaps in learning was projected; "*Gaps in learning depending on parental support*" (deputy headteacher). Another headteacher affirmed that for these pupils, "*their home life has provided limited stimulation*". A significant concern was the regression of key skills and knowledge during the period of school closures based on staff observations during remote teaching; "*Many not accessing learning and falling behind or regressing*" (reception support staff), and "*their motivation has reduced. Key skills deteriorated. Knowledge lost*" (year 3 teacher). As a result, concerns for the face-to-face return to school included; "*children's wellbeing, gaps in learning—especially between groups of learners who have engaged and those who haven't with home learning*" (year 4 teacher). Increasing inequalities were also highlighted regarding children with ALN as "*some have had no learning as have either not the support at home to help or have*

*refused to work outside of the school setting. . .list is endless working at a SEN* [*special educational needs*] *school*" (teaching assistant) and "*The vulnerable and ALN pupils generally have not engaged in the sessions*" (headteacher).

Therefore, some teachers recommended that there needed to be "*more accountability from certain parents*" (year 4 teacher) and others discussed this in relation to parental engagement, stating the importance of "*finding a way of stressing to disengaged parents the importance of completing some work at home*" (reception teacher). Recommendations to address this included "*parents being required to check in weekly*" (year 1 teacher).

However, many school staff discussed the various challenges and barriers to parents in facilitating home learning. This included parental working commitments of "key worker" occupations, recognised as those that are critical to the COVID-19 response such as roles in health and social care; "*Children of critical care workers have had limited support from parents re learning due to them working very long hours*" (headteacher). Other challenges highlighted were the number of children in the household, and parents with other working responsibilities that did not classify as a "key worker"; "*Parents who are still working (not eligible for hub) find it difficult to teach pupils and work particularly if there are two or more children at home*" (year 3 teacher). Parental language barriers were also highlighted, for example for those pupils "*who have parents with little or no English*" (teacher). School staff also found that some "*parents find teaching the work difficult*" (year 1 teacher) and suggested additional challenges for children with ALN such as "*stress for parents/carers as I teach children with ALN*" (teacher), and more broadly "*living with stressed parents*" (year 5 teacher).

Staff recommended that "*parents need to be supported more to complete the home learning*" (headteacher). To address this, suggestions included "*parents had access to a local helpline that could direct them for support*" (headteacher) and regular meetings and live feeds with teachers to access support; "*Possibly meetings with parents, to offer support and guidance in regards to best way to support their child at home. Opportunities for parents to access live feeds with teachers—if only to reassure and encourage*" (year 4 teacher). In addition, a more universal approach to supporting parents and families was suggested; "*Weekly check in with those families who need support, not just those on the vulnerable list*" (year 6 teacher). Other challenges relating to parental engagement encompassed barriers regarding digital support will be outlined in the following recommendation.

School staff also recommended working with parents regarding their expectations and suggested asking "*parents' views about home learning*" to ensure that parental expectations aligned with school priorities. In addition, primary school staff highlighted the importance of building rapport with parents to ensure "*better home/school links set up beforehand*" (year 1 teacher). Others recommended more direct communication between teachers and parents "*rather than just headteacher to parents*" (reception teacher) and emphasised that communication with parents should be an ongoing process.

## Enhance (pupil, parent and staff) digital competence through increased access and support

During lockdown and school closures, teaching was primarily delivered through digital methods requiring children to engage in remote learning online. Also associated with children's learning progression and gaps in learning, this recommendation theme covers a range of digital barriers. Pupils experienced digital exclusion through poor/inconsistent connectivity and thus a lack of access to online learning materials and digital equipment (e.g. internet access, laptop) and competing demands for equipment in the household [43]. Staff in this study also highlighted that digital competency was an issue for pupils, teachers and parents, with a lack of

digital understanding hindering pupils' ability to learn, teachers' confidence and ability to teach online and parents' ability to support their child with online home learning. Whilst staff suggested the provision of digital equipment and internet access, they emphasised that this should be matched with digital training for all, including teacher training for blended learning and training to support children and parents accessing online learning materials. This will be discussed in more detail below.

Benefits were noted by some school staff in relation to improved skills; "*I believe that many have developed their ICT skills with online resources regularly being used*" (year 2 teacher), and this was supported in instances of high parental engagement; "*For those engaged, IT skills for parents and children improved*" (reception teacher).

However, staff highlighted digital barriers to learning progression including access to technology and a general understanding and competence of basic digital skills; "*some children were unable to access online learning*" (year 2 teaching assistant) and "*lots of children who were digitally excluded suffered as the extra digital resources were not made available to the children until weeks into lockdown*" (year 6 teacher). Competing demands for use of digital technology within the household prevented children from learning; "*Lack of IT equipment where same members of the family use the equipment*" (teacher), "*some large families are having to share one device*" (year 3 teacher) and "*some children were working from phones*" (year 4 teacher).

School staff perceived that lower parental digital competence and a lack of access to digital equipment was contributing to widening learning gaps; "*Widened gap of those that have/have not devices and know-how. Lack of parental knowledge of technology has hindered some*" (deputy headteacher). Staff experiences during remote teaching and learning included "*some parents have been reluctant to use Hwb [national virtual learning environment and resource repository] as they are not familiar with it*" (year 2 teacher) and "*some parents were too embarrassed to ask for help in regards to support with technology*" (year 4 teacher).

Therefore, access to digital equipment was a consistent recommendation in the case of a blended learning approach; "*all pupils need to be given a laptop/internet access if they don't already have one. Lessons introductions should be recorded*" (year 6 teacher) and "*ensure all children are equipped with digital equipment to access online learning*" (year 3 teacher). Gathering information on children's access to digital equipment was suggested through a "*survey of access for pupils regarding IT access and internet access*" (teacher). In addition to the provision of equipment, school staff highlighted that digital training may be required for pupils; "*Children need to be trained up on how to access digital resources and live lessons beforehand*" (year 3 teacher).

Schools attempted to support those pupils without equipment through the provision of resources; "*Some pupils didn't have digital devices but this was dealt with by loaning Chrome Books*" (Year 2 teaching assistant). However, where this was not possible, a recommendation involved the combination of both digital and paper-based options for children without access to appropriate technology; "*Paper based work for pupils who cannot access online work*" (year 5 teacher).

Additional digital training for teachers was also advocated as barriers during remote teaching included "*staff confidence in use of technology and the pitfalls of live streaming*" (headteacher). Therefore, digital support was recommended; "*teachers trained to be able to deliver online teaching*" (reception teacher), and references to blended learning were also made; "*ensure that teachers have training in blended learning and that the approach is consistent across the school*" (year 6 teacher).

Regarding internet access in the home, a teacher recommended to "e*nsure all parents have access to the Internet for online learning*" (year 2 teacher). However, internet access without digital skills or competence was a barrier to parents. Therefore, supporting parents with digital

training was also advocated, with suggestions such as "*have sessions with parents on how to use Google Classroom to ensure they can support learning*" (year 4 teacher), and ensuring "*a support process in place for those parents who encounter any online difficulties*" (higher level teaching assistant).

## Adapt the learning environment and teaching practice

This recommendation theme represents school staff experiences during the phased reopening of schools, including adaptations to regular teaching practice (e.g. smaller class sizes) and the learning environment (e.g. higher staff to pupil ratio) that benefitted pupils. Staff in this study suggested that decreasing class sizes and increasing staff numbers would support whole-school wellbeing and allow learning support targeted to need. However, staff acknowledged that this required significant government investment and support to facilitate longer term adaptations to education provision.

During the summer term 2020, the guidance on reduced pupil attendance during this period enabled smaller class sizes and many reflected on the positives of this; "*The children have thrived in smaller groups. We have seen quieter pupils come out of their shells as they have not been 'drowned out' by the more enthusiastic. They have also had more individual support. We have seen a number of pupils learning more due to being in a smaller group*" (headteacher).

The reduced numbers in class during the phased return to school allowed for a focus to be placed on whole-school wellbeing, addressing previous concerns and recommendations regarding prioritising wellbeing. In addition, it allowed extra support for pupils that required help; "*Smaller classes have provided more quality 'chats' with teacher which has benefitted the children's wellbeing and been particularly helpful in reducing levels of anxiety. The more relaxed structure has had positive effects on staff and pupil wellbeing on the whole*" (year 6 support staff).

Reduced class sizes and staffing capacity was discussed in relation to the ratio of staff to pupils and being able to provide additional support where required as a result of widened gaps in learning; "*Having full classes and not having the time to talk to the pupils who will really need it. Pressure to help those pupils who haven't engaged in any learning to catch up*" (year 3 teacher).

Recommendations for the longer-term changes in class size were relayed and the benefits this could bring to pupils' learning; "*Smaller class sizes and more staff—the reason why the private sector do so well. . .The pupils have said how it is easier to learn within a smaller group*" (headteacher). However, suggestions for smaller class sizes longer term included the need for more staff and funding; "*Support in regards to mental health and wellbeing, possibly reduced class sizes, extra support for pastoral care and to help catch up. More funding*" (year 4 teacher)

Funding and investment by the government requiring long term commitment to support adaptations to current education practice was recommended. This would enable schools in reducing class sizes, especially in the context of major national curriculum reform in Wales that was published in January 2020, to start from September 2022 [44]. This included ensuring the necessary support and capacity for pupils was available; "*I understand that we have to get the economy going but if the Government wants children to 'catch up' they need to understand the positive impact of being in smaller groups. Due to budget constraints we have had large classes to keep jobs—now is the time to INVEST in learning—class sizes no greater than 20. This will have a massive impact on pupil learning*" (headteacher). Extra staffing was also suggested in relation to school autonomy, with a teacher suggesting the need for "*funding for extra staff to manage the needs of the individual school*" (year 5 teacher).

Finally, concerns relating to expectations about the enhanced cleaning and hygiene practices required in the face-to-face return to school were also remarked upon; "*Maintaining good*

*hand hygiene with a full class of 30 will be very difficult to monitor and very time consuming*" (year 5 teacher). Teachers were concerned that cleaning time will be time taken away from supporting children with learning; "*Staff regularly cleaning surfaces and toilets and won't have much time to support children*" (headteacher). However, embedding effective cleaning practice was recommended; "*I think we need to try it and make it as safe as we possibly can by making sure we have good hygiene practice in place*" (reception teacher).

### Clear communication of guidance and expectations

This final theme constitutes the importance of clear and regular communication between government and schools, the possibility of schools receiving advance notice of changes in guidance to support planning and expectations, and the challenges of incorporating generic guidance to fit within the contextual school differences and needs.

Regarding a face-to-face return to education, school staff advocated for advance notice from government and public health officials regarding any new changes in guidance to schools; "*clear guidance from day one*" (headteacher) and "*clear communications between Government, local authorities and schools*" (reception teacher). Clear guidance was a consistent recommendation by school staff who called for "*proactive communication from Government*" (year 6 teacher) and "*precise guidelines from the Government*" (headteacher).

Others called for an understanding of school needs and contexts, highlighting a need for "*clear guidance from Government with an understanding of how schools operate*" (reception teacher). School staff recommended that it was essential that communication between government and schools was done so transparently and on a regular and rapid basis; "*More information being given to schools explaining the reasoning behind decisions made, given to us before the general public rather than at the same time*" (year 3 teacher). However, a mistrust in statistics and scientific evidence was highlighted, with a learning support teacher stating there was a "*lack of confidence in the data we are given*".

Involving schools in guidance changes was valued as important, suggesting to "*Consult teachers before making decisions and let schools know their plans before announcements to the public*" (year 3 teacher) and "*knowing earlier so we can prepare*" (support staff). In particular, clear guidance and communication was highlighted consistently in relation to providing schools with adequate preparation time "*to have time to implement and understand it. Ask questions if needed*" (year 3 teacher).

Schools required regular government communication to include information on expectations; "*Regular updates for parents as well as schools on expectations of pupils with regards to Government guidelines*" (reception teacher). School staff also suggested that improved governmental communication would facilitate school-to-parent dialogue; "*Better communication with schools from the Government so that we could pass on information to parents more effectively*" (deputy headteacher), and help to manage parental expectations; "*Clear guidance and expectations from the government in order to make plans and inform parents of what school will look like. . .We cannot plan unless we have been told what it needs to look like*" (year 1 teacher).

School staff leadership advocated for a "*flexible curriculum. Trust teachers' professional judgement of their children*" (deputy headteacher) and "*accepting our school autonomy, trust us, let us innovate*" (headteacher). Strong leadership by senior management was highlighted as important by school staff, with benefits to teacher confidence and trust associated with a supportive headteacher; "*I'm also certain that my headteacher will be very supportive in the process as she always is and things will be made easier with her guidance*" (higher level teaching assistant).

Regarding a blended learning approach, teachers advocated for more "*consistent expectations of how teachers plan for home learning by their school*" (year 3 teacher). Generally,

references to consistency encompassed a school's approach, expectations and organisation to ensure that there is "*a plan in place if this happens again to make sure all pupils are involved in keeping up with their learning*" (reception teacher).

## Discussion

The COVID-19 pandemic resulted in the implementation of national school closures in more than 190 countries and impacting over 1.6 billion pupils globally [1]. Limited evidence exists to support school closures [19] as children under the age of 14 appear to have lower susceptibility to infection [2–6]. Closing schools also contributes towards widening inequalities and inequities, and negatively impacts children's physical and mental health, educational attainment, and social and economic outcomes [8–10, 14, 15]. Globally, widening inequalities in education have been reported in the UK [15], across Europe [16, 17] and the USA [18], and it is estimated that school closures result in an average loss of between 0.3 and 0.9 quality-adjusted years of schooling and a reduction in potential earnings during adulthood [11].

The reopening of schools requires research to improve understanding of the impact of school closures on the education, health and wellbeing of pupils and school staff. Furthermore, this needs to acknowledge the challenges posed during reopening and develop strategies based on "*actionable findings*" as to how nations and jurisdictions can return to a "*new (ab)normal*", informing emerging policy and practice [24, 45]. This study uses an interpretivist grounded theory approach to propose five recommendations based on the perspectives, experiences and suggestions of primary school staff about the impact of school closures during the period of March to June 2020 and the initial phased reopening of schools in the summer term 2020. These recommendations aim to identify priorities to inform emerging policy and practice, and develop strategies in post-COVID education provision that address the impacts of school closures.

### Recommendation 1: Prioritise the health and wellbeing of pupils and staff

Of great importance to primary school staff during lockdown and in considering potential future school closures was the health and wellbeing of the school population. This mirrors findings from a UNICEF return to school survey in which the most common priority for schools reopening was prioritising children's mental health and wellbeing [46]. Perceptions of school staff within this study highlighted the negative impact of school closures on children's health and wellbeing including less physical activity, increased sedentary time and screen time, social isolation and a lack of routine. This was reflected in staff observations of some pupils' weight gain, lethargy, anxiety, low mood and social disconnection upon the phased reopening of schools in the summer term of 2020.

Regarding physical health behaviours, the lockdown restrictions and 'Stay at Home' messages are likely to have disrupted children's physically active behaviours. Similar findings regarding lower physical activity levels and increased sedentary time reported in this study have also been reported in the UK [47] and globally [48] during the lockdown period. Experts have highlighted the risk of COVID-19 and impaired sleep in children through a number of potential mechanisms including stress, increased screen time and sedentary behaviours, and disturbed routines such as bedtimes and wake times [49, 50], providing possible explanations for school staff perceptions within this study. However, concerns reported in this study regarding increased screen time and social media use must also be balanced against the need for children to feel socially connected to friends [51], particularly in relation to staff observations of pupils' social isolation and disconnection upon the phased return to school.

Experts have advocated for opportunities of increased play provision without social distancing within primary schools in order to address concerns over social isolation and loneliness [52]. Within this study, some schools offered increased provision of outdoor learning and play during the phased return in order to prioritise pupil health and wellbeing [53]. However, some staff in this study suggested that additional staff support and infrastructure was required for outdoor learning provision in the face-to-face return to education. For outdoor learning to be embedded within the school curriculum, it must be valued by education inspectorates as a feasible method of achieving curricular aims [53], and the new Curriculum for Wales may offer opportunities for this longer term [44].

Protecting and promoting the mental health and wellbeing of children has been a key global public health focus prior to COVID-19 [54], and the pandemic and subsequent school closures brings with it new challenges in addressing this [55]. Whist this study and others [56] suggests some benefits to children's wellbeing from increased time spent with family and strengthened family relationships during the lockdown period, school staff also observed anxiety amongst pupils returning to school. Elsewhere, parent-carer reported information collected during the period of school closures suggests increases in primary school children's emotional, behavioural and attentional difficulties [57], and that worries about missing school was the most common concern of pandemic-related anxiety for children [50]. The longer term risks to children's mental health must also be considered as evidence has highlighted associations between loneliness and episodes of depression, with worse outcomes in scenarios of forced isolation such as school closures and quarantine measures [58]. Much of this research however, is based on parent or staff perceptions of how children's health has been affected and it will also be important to consider research regarding objective measures and accounts from children themselves [59–61]. Furthermore, for children's mental health to be considered a key priority within education provision as highlighted in recommendations in this study and advocated for by UNICEF [62], this must also be reflected within wider education and public health policy and investment.

Recommendations by school staff in this study were made to remove the pressure placed on assessments and attainment and rather, focus on children's wellbeing. This finding has been brought up in previous research [63] but has again been brought to the forefront of attention in the context of the current situation. This is of particular relevance in Wales, as the Welsh Government published the major reforms to the national curriculum in January 2020, to phase in for learners from September 2022 [44]. Linking back to a number of themes identified in this study, one of the six "*areas of learning and experience*" in the new Curriculum for Wales is "*Health & Well-being*", which encompasses "*physical health and development, mental health, and emotional and social well-being*", increasing the prominence and profile of this topic within schools [64]. In a post-COVID education landscape, perhaps opportunities for the integration of health and wellbeing within curriculum reform demonstrated within Wales will be replicated and adopted more widely, thus, reflecting a true synergy between health, wellbeing and education given their interconnectedness [65].

Staff in this study also advocated for teacher wellbeing to be considered and prioritised, conveying the difficulty in maintaining a work/life balance during lockdown due to increased pressure, higher workloads and personal challenges such as childcare. These challenges and wider impacts have been recognised across all educational settings, including higher education [66, 67]. A survey examining teacher wellbeing during lockdown highlighted that primary school teachers reported lower wellbeing than secondary teachers, and lower levels further displayed by those in a senior leadership positions [68]. These findings, and those reported in this study therefore highlight the importance of prioritising not just pupil wellbeing, but also that of school staff during future scenarios of blended learning and within the return to face-to-

face education. Within this study, staff isolation due to social distancing requirements and a lack of face-to-face interaction with colleagues was highlighted, and a recommendation by school staff was protecting their staff breaks to promote wellbeing. Indeed, Roffey highlights that research regarding teacher wellbeing is often focused on factors that lead to stress rather than those factors promoting positive wellbeing [69]. Whilst performance management for teachers remains a statutory requirement, perhaps incorporating measures and targets that foster positive wellbeing for staff with input from staff themselves would be welcomed by schools. This is of particular importance for schools as teacher wellbeing is considered a critical factor in creating stable environments for children to thrive [69], and has also been associated with academic achievement [70].

Therefore, recommendations from this study and evidence within the wider literature demonstrating the range of negative impacts of school closures on children support global calls for the prioritisation of health and wellbeing in both the short and long term provision of education and within emerging policy design [71].

## Recommendation 2: Focus on enabling parental support and engagement

Another key recommendation by primary school staff was to focus on enabling parental engagement due to wide variations in home learning provision during lockdown and concerns about the increasing gap between children's learning progression. The COVID-19 pandemic has exposed the persisting inequalities that exist within society and span across multiple domains including socio-economic indices, health and education [72]. Widening education inequalities have been reported in other studies in the UK [15], Europe [16, 17] and the USA [18]. Research suggests that markers of home learning are socially patterned, with the most favourable profiles of home learning observed in the highest family income group [73]. This has also been evidenced in the COVID-19 pandemic, with the Institute for Fiscal Studies (IFS) demonstrating that children from better-off families were spending 30% more time on home learning activities than those from poorer families [74]. Although not measured in this study, it is possible the same socio-economic influences are operating, however it is also clear that the COVID-19 pandemic has generated challenges for all families regardless of socio-economic background. In addressing gaps in learning progression as exposed in this and other studies, Governments worldwide are responding with education recovery packages aiming to support children to "*catch up on lost learning*" [75–77]. However, it is important to also consider the recommendation of prioritising the health and wellbeing of pupils and staff and ensure that policy and practice efforts are not at the detriment to health and wellbeing.

Caution is required in interpreting the findings relating to parental engagement as they may not necessarily be reflective of disengaged parents but rather, of wider barriers that exist. Additional challenges for all families will influence a parent's ability to support their child's learning, including working commitments or parental health issues. School staff in this study recognised a number of other barriers to parents engaging with home learning activities, including parents finding the work difficult to teach, having a number of children within the household, parental stress, English as a second language, challenges for Welsh-medium education and children with additional learning needs. Although parents were not surveyed in the current study, research conducted in Ireland with parents of primary school-aged children highlighted that a lack of familiarity with the curriculum, a lack of time and high levels of stress impacted parental confidence in home learning support [78]. School staff in the current study recommended the use of helplines for parents, live feeds with teachers where parents could access support and weekly check ins with parents to ensure parents felt confident in helping their child with home learning. Regular communication and building rapport between class

teachers and parents was suggested as important, and helps to build trust and understanding at a time of great uncertainty. Indeed, the return to school requires a high level of communication between schools and parents in relation to school practice and wider school operational factors. Preliminary findings from the UNESCO school reopening survey highlighted that good communication between schools, parents, local authorities and wider stakeholders is key to the effective reopening of schools and a return to education [79].

## Recommendation 3: Enhance (pupil, parent and staff) digital competence through increased access and support

A key recommendation highlighted by school staff in this study was the importance of providing digital support and training to pupils, teachers and parents. Again, this has particular relevance with the emerging Curriculum for Wales reforms, which enshrines digital competence as a statutory cross-curricular skill alongside literacy and numeracy for all learners from ages 3–16 [80]. Whilst some improvement to children's ICT skills were noted by staff, remote learning brought with it a number of barriers for children and is likely to contribute to the widening gaps in learning reported by primary school staff [81], alongside increased awareness of digital/data poverty and the resulting longer-term impact of digital exclusion [9]. Reasons for digital exclusion cited in this study included competing demands for the use of equipment within the household, lack of access to digital equipment or internet, and low parental digital competence. A lack of digital resources at home and online has previously been highlighted as a barrier to parents supporting their children with home learning [78]. Therefore, it is unsurprising that a consistent recommendation by school staff in this study was the provision of equipment such as laptops. Staff suggested using surveys to gather information on pupils requiring equipment. In addition, others recommended that work set for periods of home learning must include paper-based options for those without access to learning online. The government ensured provision of digital devices to children without access during the lockdown period. However, school staff also suggested that offering digital training and support to parents was important, such as information on how to use the various resources provided by schools in order to support parents with online learning activities with their child. Therefore, equipment provision alone may be unlikely to provide solutions to these barriers without the inclusion of appropriate training and support for increasing digital competence.

Low teacher confidence in delivering lessons virtually was also highlighted in this study, with recommendations to address this including providing staff training for remote teaching, supporting recommendations in a recent evidence assessment by the Education Endowment Foundation [81]. Over half of teachers surveyed in England during lockdown reported to find online teaching stressful. Primary teachers were the least likely to have experience in delivering online lessons and thus were the least prepared for digital teaching [68]. Notably, a report by the Organisation for Economic Cooperation and Development (OECD) demonstrated inequalities in digital competence among staff teaching in deprived schools when compared with more affluent schools, with children from more affluent schools having greater access to teachers with greater digital competence than those at more deprived schools [82]. This may serve to further increase inequalities in learning outcomes between the most deprived and affluent children. Looking to possible future scenarios of a blended learning approach, it is essential that the current teaching workforce are equipped with the necessary skills and training and feel confident and supported in delivering teaching online. Furthermore, whilst COVID-19 has exposed that digital barriers to learning and teaching are present even within the increasing digitalisation of education, perhaps this will be the catalyst for a renewed focus

on ensuring the necessary training and infrastructure are in place for future learners, families and the teaching workforce [22].

## Recommendation 4: Adapt the learning environment and teaching practice

A longer-term commitment by government was recommended to create a fundamental shift in education practice relating to class sizes, staffing and funding in supporting children impacted by school closures. This would enable teachers and teaching assistants to provide direct educational and pastoral support to children identified as requiring additional help in relation to the widening gap in children's learning progression during school closures. During the phased reopening of schools in which attendance was capped, the smaller classes and more direct support provided to pupils was recognised as a benefit by school staff. However, school staff in this study recognised that providing this in face-to-face education would require additional staffing, in addition to significant funding and investment from government. We have seen recent strategic education investments by the Welsh and Scottish governments alongside their curriculum reform initiatives, however it remains unclear whether these investments will support the possibility of sustainable smaller class sizes [77, 83]. Another benefit to smaller class sizes was that it would help schools in maintaining social distancing and good hygiene practice. School staff in this study were concerned that maintaining good hygiene would be a challenge within full classes, in addition to taking time away from learning or other activities. However, it was acknowledged that this practice was paramount for the safety of staff, pupils and families and is an integral component of this 'new normal'.

## Recommendation 5: Clear communication of guidance and expectations

Communication, clear guidance and managing wider expectations from government were also endorsed within this study. Regarding the need for governments to recognise school needs and contexts identified in this study, the NFER suggest tailored guidance should be shared based on school needs and demographics. This would also support schools given their varying approaches during school closures, the phased reopening and a face-to-face return to education.

The challenges of devolved government responsibilities and associated media coverage could account for these findings, in addition to the variation in guidelines and regulations across the four home nations. School staff advocated for receiving regular and clear guidance and advance notice of changes in guidance to allow preparation time, to facilitate communication with parents and to enable schools to manage parental expectations more effectively. These findings are mirrored within other research examining the implementation of preventive school-based measures in primary schools in England, with participants commenting on the short notice of government guidance and school staff advocating for greater clarity and the involvement of schools [84]. Whilst it is likely that this is also relevant across different sectors impacted by changes to working and operation, it is important to acknowledge the practicalities associated with a rapidly changing pandemic and subsequent governmental decision making. Decisions regarding education delivery including those relating to school closures, reopening or adaptations to practice are often reactive to rapidly emerging data and thus, communicating changes is most efficient through national media coverage.

The challenge of communicating latest evidence and changes to guidance to schools could account for the feelings of political mistrust by school staff conveyed in this study and others [85], with staff in the 'Back to School' report also feeling responsible for leading on decision making [86]. School staff in this study identified strong leadership as important and, associated positive examples of a supportive headteacher with improved teacher confidence and trust. It

is clear that the COVID-19 pandemic has caused great disruption to the education system, requiring school leaders to be responsive and adaptive to rapidly changing guidance and scenarios. Whilst there is a lack of research examining how school leaders are responding to the pandemic, a paper by Harris and Jones offer several insights that have emerged, and state that trust within the leadership system is essential for collective action during these unprecedented times [87].

## Strengths and limitations

Findings from this study provide detailed experiences of a range of primary school staff about the period of school closures due to COVID-19, their perspectives about the phased face-to-face reopening of schools and recommendations for the future. This study contributes to the limited evidence base that currently exists regarding the impact of school closures on pupils and school staff, and the challenges and opportunities posed for schools during a return to face-to-face education. Importantly, the recommendations that conclude this study offer practical insight and learning to support schools returning to in-school education. In addition, the findings and recommendations from this study provide knowledge on priority areas for future policy focus in reducing the inequalities that have widened during the pandemic.

There are limitations to review when considering the findings from this study. Although all schools in Wales were contacted with details regarding the staff survey, the findings in this study only represent a convenience sample of those that participated. The perspectives captured in this study may not account for the full breadth of lived experiences of all primary school staff during school closures and face-to-face reopening. It is possible that those that participated in this study work within schools that are 'research engaged'. Those that did not participate may have different experiences and perspectives during this period to those reported in this study. Schools adopted varying approaches to remote teaching and in their return to education and thus, those reflected in this study may not encapsulate all approaches. However, the sample consists of a diverse range of primary school staff including headteachers, teachers and support staff working across all year groups of primary school. School staff in this study worked in both urban and rural schools from diverse socio-economic communities within 16 of 22 local authorities in Wales. The experiences of primary school staff are based on their perceptions of the impact of lockdown on pupils and quantitative assessments of reported concerns were not undertaken in this study. It is also important to consider the unprecedented adaptations to education provision and higher working demands in contributing towards school staff stress and the impacts this may have on their perceptions and experiences, or ability to participate in the study.

## Conclusion

In summary, primary school staff perceive that gaps in learning, health and wellbeing have increased and inequalities have widened during school closures. Schools are now tasked with closing this gap, whilst protecting and safeguarding children. It is essential for research to focus on the mechanisms that optimise and improve outcomes for children both during and after the COVID-19 pandemic, and translate these as "*actionable findings*" to inform emerging policy and practice [21, 24]. This work, along with related findings, has contributed to emerging policy and practice at the national scale in Wales, offering policymakers an evidence base for school-based practice into 2021 and beyond. Furthermore, research must support and develop strategies for education provision post-COVID that address these impacts of school closures. Therefore, the recommendations for current and future education provision in this study based on primary school staff perspectives and experiences during school closures and

the initial phased face-to-face reopening of schools are: (i) prioritise the health and wellbeing of both pupils and staff; this includes more focus on wellbeing activities and less focus on attainment and assessments during the return to school and protecting staff breaks to promote workplace wellbeing; (ii) focus on enabling effective parental engagement and explore all avenues to increase parental engagement including introducing support sessions, helplines and regular check ins during periods of home learning; (iii) build digital competence amongst pupils, teachers and parents; for example, by ensuring both the provision of equipment and training in building digital skills and offer a combination of paper-based and digital home learning activities; (iv) consider opportunity for smaller class sizes and additional staffing which will ensure support is directed to need whilst providing pastoral care; and (v) improve the mechanism of communication between schools and families, and between government and schools. The broad nature of these recommendations cover the wide ranging impacts of school closures. This calls for future mixed-methods research to explore each topic in greater detail to examine and quantify the depths of these impacts and challenges, particularly since the return of face-to-face education provision.

## Supporting information

**S1 File. Primary school staff survey.**
(DOCX)

## Acknowledgments

The authors would like to thank all participating primary schools and school staff that took part in this study. Infrastructure support was received by the National Centre for Population Health and Wellbeing Research.

## Author Contributions

**Conceptualization:** Emily Marchant, Sinead Brophy.

**Data curation:** Emily Marchant, Charlotte Todd, Michaela James.

**Formal analysis:** Emily Marchant, Charlotte Todd.

**Methodology:** Sinead Brophy.

**Project administration:** Emily Marchant, Charlotte Todd, Michaela James.

**Supervision:** Sinead Brophy.

**Writing – original draft:** Emily Marchant, Charlotte Todd.

**Writing – review & editing:** Emily Marchant, Charlotte Todd, Michaela James, Tom Crick, Russell Dwyer, Sinead Brophy.

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
