## [Decision Letter · Decision Letter 0]

2 Feb 2021

PONE-D-20-34984

Primary school staff reflections on school closures due to COVID-19 and recommendations for the future: a national qualitative survey

PLOS ONE

Dear Dr. Marchant,

Thank you for submitting your manuscript to PLOS ONE. After careful consideration, we feel that it has merit but does not fully meet PLOS ONE’s publication criteria as it currently stands. Therefore, we invite you to submit a revised version of the manuscript that addresses the points raised during the review process.

We look forward to receiving your revised manuscript.

Kind regards,

Amanda A. Webster

Academic Editor

PLOS ONE

Journal Requirements:

2.Please provide additional details regarding participant consent. In the ethics statement in the Methods and online submission information, please ensure that you have specified what type you obtained (for instance, written or verbal, and if verbal, how it was documented and witnessed). If your study included minors, state whether you obtained consent from parents or guardians. If the need for consent was waived by the ethics committee, please include this information.

3. Please include additional information regarding the survey or questionnaire used in the study and ensure that you have provided sufficient details that others could replicate the analyses. For instance, if you developed a questionnaire as part of this study and it is not under a copyright more restrictive than CC-BY, please include a copy, in both the original language and English, as Supporting Information, or include a citation if it has been published previously.

4. In the Methods, please discuss whether and how the questionnaire was validated and/or pre-tested. If these did not occur, please provide the rationale for not doing so.

5.We note that you have indicated that data from this study are available upon request. PLOS only allows data to be available upon request if there are legal or ethical restrictions on sharing data publicly. For information on unacceptable data access restrictions, please see http://journals.plos.org/plosone/s/data-availability#loc-unacceptable-data-access-restrictions.

6.We note that the grant information you provided in the ‘Funding Information’ and ‘Financial Disclosure’ sections do not match.

Reviewers' comments:

Reviewer's Responses to Questions

**Comments to the Author**

1. Is the manuscript technically sound, and do the data support the conclusions?

Reviewer #1: Partly

Reviewer #2: Yes

2. Has the statistical analysis been performed appropriately and rigorously? 

Reviewer #1: N/A

Reviewer #2: N/A

3. Have the authors made all data underlying the findings in their manuscript fully available?

Reviewer #1: Yes

Reviewer #2: No

4. Is the manuscript presented in an intelligible fashion and written in standard English?

Reviewer #1: Yes

Reviewer #2: Yes

5. Review Comments to the Author

Reviewer #1: This is an interesting study and certainly an import area of interest. My suggestions are as follows:

- You have provide some current literature –perhaps include some studies from other countries for an international audience

- You have talked about negative consequences and inequalities were there any positive outcomes and opportunities

- You are covering a lot of areas---child development, wellbeing, health , economic, educational outcomes ---perhaps needs to narrow the focus

- You needed to provide more information and justification on the survey development and trial- trustworthiness’ and reliability??

- You jumped very quickly from data coding to recommendations I felt there was a missing step –the analysis

- I would of liked to see some theoretical underpinning to inform coding/analysis and discussion

- Recommendation and future research can go at the end of the discussion

- Some good ideas in the discussion but need to show more analysis and less repetition of the findings –the SO What?

- Perhaps a figure to describe your findings and then less detail in the findings sections

- Don’t use recommendations in the abstract or findings –reduce to key ideas/categories –convert the recommendations into implications and unpack in the discussion

- I would narrow the focus you are trying to cover too many areas e.g., wellbeing. Or inequalities opportunities

- Need to explain the relevance of FSM for international audience

- Don’t include too much ethics detail –sought and approved is enough

- At beginning of discussion re-state key literature this was missing

- Put survey questions in appendix not in text

You have some interesting data and I wish you all the best with you next dr

Reviewer #2: Thank you for inviting me to review this article involving a qualitative survey with primary school staff on their experiences around school closures due to COVID-19 and recommendations for the future. I thought it was a very informative paper, with extremely timely and potentially helpful recommendations. My main feedback is about ensuring the rigour of the methodology and analysis in order to feel confident that the recommendations are grounded in the data.

More specifically, my feedback is as follows:

Title: the use of the terms ‘national qualitative’ may be a bit of a conflict as by calling it ‘national’ implies representativeness, which qualitative work does not seek to do. It may be more helpful to specify that it was conducted in Wales within the title.

Introduction

• The first paragraph was extremely long and it was difficult to follow the line of argument – could they break this paragraph up a bit?

• Could the authors explain what the “continuity of learning” policy was and what it involved in practice for school staff and families?

• The tense changed from past to current

Method

• The survey was distributed in July but teachers were described as reporting on the reo-opening of schools in September (line 89) – could the authors provide dates of when the survey was completed?

• What were the inclusion/exclusion criteria for taking part?

• Although the sampling method was described as purposive, it appeared that it was based more on convenience sampling (i.e., all schools in Wales were contacted). To what extent did they manage to capture important perspectives from difficult to reach people?

• Attempts to recruit participants and reasons for non-participation should be stated.

• Second coding was described as being for ‘accuracy’; however, qualitative research wouldn’t typically expect ‘accuracy’ and instead would recognise there is no one correct way to interpret data.

• The theoretical framework for the analysis was not clearly explained and there were no references related to the methodology.

• It would be good to justify the use of this analysis for survey response data.

• Given that qualitative researchers closely engage with the research process, it would be helpful to know more about the research team and provide evidence of reflexivity in the process of analysis.

• Did researcher use memo-ing in order to demonstrated how they developed their understanding of the data?

• Did the team obtain feedback from participants on the research findings to validity to the team’s interpretations?

Results

• The results were framed as recommendations for the future so I wonder whether the research questions (that related to experience of school closures as well as recommendations) need to be a bit clear that the focus is on using the experiences to generate recommendations.

• With some of the quotes, it wasn’t clear to what extent they reflected staffs’ experiences versus their expectations, e.g, “the children will have been affected by being out of school for

• 186 so long, and will have trouble re-adjusting to school life” (line 186). I would be cautious about interpreting this as young people having had trouble adjusting – but more than staff expected this to be the case.

• There were other quotes where it wasn’t always clear that this reflected the theme, e.g., “a lot of time spent on computers and not going outside” (year 2 teaching assistant) (line 199) may just reflect lockdown and may not mean the child requires anything other than coming out of lockdown.

• It would be helpful to better understand how the quotes about gaps in learning (p216) fitted with the recommendation around a focus on wellbeing – rather than a focus on maintaining/improving learning during lockdown

• Results were very much at the summary level, but given the purposive sampling, I was interested to know whether there was variability in experiences/perspectives on the basis of the sampling characteristics

Discussion

• Whilst it would be ideal for schools to receive advance notice around changes in guidance, it may be helpful to recognise the ever-changing nature of the pandemic – so although ideal to ‘ensure schools receive advance notice of changes in guidance’ this may not always be possible.

6. PLOS authors have the option to publish the peer review history of their article (what does this mean?). If published, this will include your full peer review and any attached files.

Reviewer #1: No

Reviewer #2: No

---

## [Author Response · Author response to Decision Letter 0]

19 Mar 2021

Reviewer #1

1.1 You have provide some current literature –perhaps include some studies from other countries for an international audience

Thank you for raising this, we agree that the inclusion of international literature would be useful for international readers. This has been addressed several times throughout the manuscript, including discussions of the impact of school closures on children in Europe and the USA. Other examples of international literature included are listed below:

• “Globally, it is estimated that school closures result in an average loss of between 0.3 and 0.9 quality-adjusted years of schooling, contributing to a reduction in potential earnings during adulthood [11].” (page 2, 42-44)

• “Widening educational inequalities have also been observed internationally including across Europe [16,17] and the USA [18].” (page 3, line 65-67)

• “Globally, widening inequalities in education have been reported in the UK [15], across Europe [16,17] and the USA [18], and it is estimated that school closures result in an average loss of between 0.3 and 0.9 quality-adjusted years of schooling and a reduction in potential earnings during adulthood[11].” (page 24, line 855-859)

• “Similar findings regarding lower physical activity levels and increased sedentary time reported in this study have also been reported in the UK [40] and globally [41] during the lockdown period.” (page 25, line 891-893)

• “Protecting and promoting the mental health and wellbeing of children has been a key global public health focus prior to COVID-19 [47]” (page 26, line 931-933)

1.2 You have talked about negative consequences and inequalities were there any positive outcomes and opportunities

That is an interesting point that you have raised. The significant majority of responses from primary school staff about the impact of school closures on children captured the wide range of negative consequences spanning the domains of learning, health, wellbeing and development. However, to address your point we have included some examples relating to some of the positive impacts relayed by staff. These include additional time with family, strengthened family relationships, ICT skills and children’s wider skill development through non-curriculum activities and practical opportunities. We now feel the balance of positive and negative impacts reflects the perspectives and experiences of primary school staff in this study. 

• “Whilst on one hand, positives to school closures were highlighted by some school staff relating to potential benefits to children’s wellbeing through strengthened family relationships and activities; “Spending time with family, baking, walking etc has been hugely beneficial to their wellbeing” (headteacher)” (page 10, line 326-328)

• “Benefits cited by school staff included children receiving one-to-one support from parents, and positive examples of parental engagement were discussed; “parents more aware of the learning needs of their children” (year 3 teacher) and a sense of “parents’ realisation of what pupils are taught” (year 3 teacher).” (page 14, line 515-519)

• “In the case of high levels of family support, school staff reported improvements to children’s wider development, enhancing skills not covered in the traditional curriculum; “Some are learning new skills not taught in primary schools” (reception teaching assistant) and through practical opportunities within the home; “Lots of children have learnt life skills such as cooking, washing and budgeting” (year 2 teaching assistant) and outdoors; “They learnt new life skills like riding bikes” (year 3 teaching assistant).” (page 15, line 540-545)

• “Experiences of high parental engagement were also reflected in improved relationships with and communication between the school and parents” (page 15, line 546-547)

• “Benefits were noted by some school staff in relation to improved skills; “I believe that many have developed their ICT skills with online resources regularly being used” (year 2 teacher), and this was supported in instances of high parental engagement; “For those engaged, IT skills for parents and children improved” (reception teacher).” (page 18, line 645-649)

• “During the summer term 2020, the guidance on reduced pupil attendance during this period enabled smaller class sizes and many reflected on the positives of this” (page 20, line 715-717) 

1.3 You are covering a lot of areas---child development, wellbeing, health , economic, educational outcomes ---perhaps needs to narrow the focus

Thank you for raising this suggestion regarding the scope of the findings. We agree that the manuscript is covering a wide range of topics, however we feel the manuscript provides a true reflection of the vast impacts of school closures on children, based on the perspectives and experiences of those in practice. This is also highlighted in many examples of global research into these impacts, including reports by UNESCO and the WHO, referenced in the manuscript. Furthermore, the economic impacts outlined in the Background section provide a longer-term view of children’s outcomes as a result of school closures and are important for the reader to consider. 

Therefore, we would like to maintain this broad focus of the impacts across the domains of child development, health and education outcomes and think this fits well within the inter-disciplinary scope of PLOS ONE. 

1.4 You needed to provide more information and justification on the survey development and trial- trustworthiness’ and reliability?? 

Thank you for highlighting this important point regarding the reporting of methods within our manuscript. We agree that we should provide more information relating to the survey development, trustworthiness and reliability. Regarding the survey development we have included additional information including dates the survey was open, the input from key stakeholders and primary school staff, and piloting to ensure wording and usability for school staff. This is discussed in the ‘School staff survey’ section and listed below. Furthermore, we have included a full copy of the survey in Supplementary File 1:

• Supplementary File 1: School staff survey

• “The survey was open for responses between 9th and 31st July 2020. Inclusion criteria for participating in the survey was any primary school staff member (headteachers, deputy headteachers, teachers, teaching assistants, support staff) working within a local authority maintained primary school in Wales, UK.” (page 5, line 148-151)

• “Rapid development of the survey was required due to limited notice on changes to education provision, including the announcement of the return to school. The development of the survey questions was based on input from the research team specialising in child health and education research (authors), key stakeholders in education (two regional education consortia curriculum staff) and two school staff from different primary schools who were independent to the research study (one teacher, one headteacher). Following revisions and refinement from key stakeholders and primary school staff, the survey was piloted with the school staff to ensure wording and usability. The final survey contained 25 questions in total, consisting of a combination of categorical questions relating to demographics (school name, role, year group), school reopening logistics (e.g. full/part-time structure, class sizes, overall attendance proportion, outdoor and play provision) and support/training requirements (e.g. professional development training and support). A range of open-ended questions were included to explore the research question (e.g. benefits/negative impacts of school closures, concerns and recommendations for full-time reopening). These questions were open-ended in order to gather information rich narratives of participants’ views, experiences and recommendations[27]. A copy of the full survey is available in the Supporting Information. For the purpose of this research study, responses relating to lockdown/school closures were gathered from questions 18, 19 and 20, and responses relating to the full-time return to education were gathered from the questions 21, 22 and 23.” (page 5-6, line 157-180)

1.5 You jumped very quickly from data coding to recommendations I felt there was a missing step –the analysis

We are grateful for this comment as you have highlighted an important point that requires further descriptions of our analysis process to enhance the manuscript’s trustworthiness. We have included additional information describing the data analysis process, following the six stage method proposed by Braun and Clarke. We have also outlined how we ensured trustworthiness in relation to your point above, including respondent validation and the use of memoing, discussed in detail in the ‘Data analysis’ section:

• “This research study adopted an interpretive approach through thematic analysis of open-ended survey responses. Thematic analysis is considered a suitable approach in gaining meaningful comprehension of patterns in participant perspectives captured using a range of qualitative methods including open-ended survey questions[31,32]. Thus, using this approach this study aimed to elicit participants’ experiences relating to school closures, the initial phased reopening of schools and their recommendations to inform the full-time return to education.

The lead (EM, PhD) and second (CT, MSc) researcher involved in data analysis were both female and had previous experience in conducting thematic analysis in qualitative school-based research. The researchers did not have any interaction with participants. Responses from the school staff survey were downloaded to an Excel spreadsheet. Participants were assigned a unique study ID and identifiable information removed. The coding process followed the six steps outlined by Braun and Clarke [32]. Firstly, the lead researcher (EM) systematically read responses several times to facilitate immersion in the data. Re-reading responses allowed familiarity with the data, and the use of memoing recorded notes of patterns and emerging insights relating to coding ideas. Thoughts relating to decision processes were documented in a reflexive journal [33,34]. These documented notes formed the basis of the second stage of coding, whereby initial codes using words and phrases were manually assigned to participants’ responses. The second researcher (CT) independently coded random survey responses from 50 participants. These coding responses were compared between researchers to account for consistencies or inconsistencies in their interpretations of responses and associated codes. Agreement between researchers was very high; if there was a discrepancy or disagreement, a third researcher adjudicated (SB). This method attempts to reduce researcher bias and enhances the validity of codes and categories assigned[35]. Following this, the third stage of coding involved the two researchers working together through an extensive process of collating, combining and categorising codes into over-arching themes and sub-theme headings. In cases where codes were merged, removed or renamed, notes were taken in a reflexive journal and raw files were saved as new versions and archived with dates in order to keep an audit trail of the coding process[34]. Throughout these processes, a codebook and manual was developed to record codes and themes. The themes and sub-themes were further refined in stage four to ensure clear distinctions between themes and then grouped into over-arching recommendations based on school staff suggestions for future school closures and the reopening of schools. The lead researcher then manually reviewed the open-ended survey responses and coded according to the final list of recommendation themes and sub-themes. These responses containing themes and sub-themes were compiled to a master copy document that was used as a reference for stages 5 and 6 whereby themes were written and discussed within the findings. 

• To ensure trustworthiness, the grouping of survey responses into recommendation themes developed by the researchers were screened by a primary school. The interpretation and discussion of themes and the final manuscript was also peer-reviewed for member checking by a primary school headteacher (study participant, RD) for corrections, clarification or confirmation [34] [36].” (page 8-9, line 231-313)

1.6 I would of liked to see some theoretical underpinning to inform coding/analysis and discussion 

Thank you for raising the importance of theoretical underpinning, we have addressed this in the texted listed above in comment 1.5, presented within the ‘Data analysis’ section, describing the six stage process proposed by Braun and Clarke to state the theoretical approach to data coding. We have also included an additional sentence to justify thematic analysis of open-ended survey questions. 

As above (point 1.5), (page 8-9, line 231-313)

1.7 Recommendation and future research can go at the end of the discussion 

Thank you for providing this suggestion. We have re-structured the discussion in order of recommendation, as this is the key outcome of our research question and the practical basis in which we wish this paper to address. We will further justify our reasoning behind focussing the findings and discussion purely on the recommendations in the other points you have raised. 

When considering your interesting comments and suggestions regarding changes to the structure of the recommendations, we felt that perhaps a re-structure of the manuscript could provide more clarity and differentiation when discussing the recommendations. 

We hope this ensures that the translation of knowledge from research to practice is enhanced, and clearly presents how we have addressed our research question of reflecting on perspectives and experiences of school staff to identify recommendations for the future in informing emerging policy and practice. 

1.8 Some good ideas in the discussion but need to show more analysis and less repetition of the findings –the SO What? 

Thank you for raising this as we want to ensure our findings can be translated within policy and practice and inform future education delivery. 

Firstly as stated above, we have re-structured the discussion section by recommendation in order to provide clarity in interpreting the findings in relation to the recommendation. This reflects the further detail we have provided in stating and addressing our research question; using the perspectives and experiences of school staff to identify recommendations, which is the overarching take home message of this manuscript. 

Secondly, we have included additional points of discussion in terms of the ‘So what’ message. For example, whilst protecting and promoting children’s mental health has been a global priority prior to COVID-19, the pandemic has brought with it additional barriers and challenges, and exposed the extent to which this is a global concern. Therefore, whilst global organisations such as UNESCO and UNICEF have advocated for children’s mental health to be considered a priority following COVID-19, this must be reflected in policy design and investment. On a wider education level, we have put forward the suggestion of other countries mirroring the integration of health and wellbeing within curriculum reform observed in Wales. In addition, we have proposed that the awareness of digital poverty as a result of COVID-19 may be a catalyst for a renewed focus on ensuring the necessary training and infrastructure are in place within increasingly digitalised education.

 Examples of additional ‘So what’ messages are outlined below: 

• “The reopening of schools requires research to improve understanding of the impact of school closures on the education, health and wellbeing of pupils and school staff. Furthermore, this needs to acknowledge the challenges posed during reopening and how countries can return to a “new (ab)normal”, informing emerging policy and practice.” (page 24, line 860-863) 

• “Experts have advocated for opportunities of increased play provision without social distancing within primary schools in order to address concerns over social isolation and loneliness [45]…… For outdoor learning to be embedded within the school curriculum, it must be valued by education inspectorates as a feasible method of achieving curricular aims[46], and the new Curriculum for Wales may offer opportunities for this longer term[37].” (page 26, line 922-930)

• “Protecting and promoting the mental health and wellbeing of children has been a key global public health focus prior to COVID-19[47], and the pandemic and subsequent school closures brings with it new challenges in addressing this[48]. Whist this study and others[49] suggests some benefits to children’s wellbeing from increased time spent with family and strengthened family relationships during the lockdown period, school staff also observed anxiety amongst pupils returning to school” (Page 26, line 930-936)

• “The longer term risks to children’s mental health must also be considered as evidence has highlighted associations between loneliness and episodes of depression, with worse outcomes in scenarios of forced isolation such as school closures and quarantine measures [51]” (Page 26, 940-944)

• “Furthermore, for children’s mental health to be considered a key priority within education provision as highlighted in recommendations in this study and advocated for by UNICEF [55], this must also be reflected within wider education and public health policy and investment.” (Page 27, line 941-944)

• “In a post-COVID education landscape, perhaps opportunities for the integration of health and wellbeing within curriculum reform demonstrated within Wales will be mirrored globally, thus, reflecting a true synergy between health, wellbeing and education given their interconnectedness[58].” (Page 947-950)

• “Therefore, recommendations from this study and evidence within the wider literature demonstrating the range of negative impacts of school closures on children support global calls for the prioritisation of health and wellbeing in both the short and long term provision of education and within emerging policy design [63].” (page 28, line 1005-1008)

• “In addressing gaps in learning progression as exposed in this and other studies, Governments worldwide are responding with education recovery packages aiming to support children to “catch up on lost learning”[67–69]. However, it is important to also consider the recommendation of prioritising the health and wellbeing of pupils and staff and ensure that policy and practice efforts are not at the detriment to health and wellbeing.” (page 29, line 1068-1072)

• “Furthermore, whilst COVID-19 has exposed that digital barriers to learning and teaching are present even within the increasing digitalisation of education, perhaps this will be the catalyst for a renewed focus on ensuring the necessary training and infrastructure are in place for future learners, families and the teaching workforce[22].” (page 32, line 1160-1163)

1.9 Perhaps a figure to describe your findings and then less detail in the findings sections

We acknowledge your point about integrating findings within a figure, however we feel that the comprehensive discussion of findings and using in-text quotes to explain key points is warranted in this qualitative study. We feel that this presentation allows the reader to be immersed in the findings and embedded quotes helps to provide a detailed story of the primary school staff perspectives and experiences, enhancing the readability and vividness of the findings. Furthermore, we have ensured that each quote supports the interpretation that we have put forward of the key topics, and the richness of this data helps to illustrate the viewpoints of school staff that we feel would be lost if removed from in-text. 

1.10 Don’t use recommendations in the abstract or findings –reduce to key ideas/categories 

Further to our points listed above, as generating a list of recommendations based on the perspectives and experiences of primary school staff is the over-arching aim, we have re-structured the paper accordingly. By presenting the recommendations clearly and succinctly in the abstract, findings and discussion, we hope to provide a manuscript that facilitates knowledge transfer between research, policy and practice. We feel that purely presenting the findings as key themes or ideas would limit the possibility of transferring learning into action. 

1.11 Convert the recommendations into implications and unpack in the discussion

We are grateful for your suggestions regarding the format of the recommendations presented in our manuscript. As this was a suggestion in a number of your comments, we considered that the manuscript required clarity and focus. To address this, we have refined our research question to make it clear that we are reflecting on staff perspectives and experiences to generate a set of recommendations for schools. We feel this is now a strength of the paper as we aim to disseminate the findings widely to primary schools and policy, providing clear points for action and impact. 

1.12 I would narrow the focus you are trying to cover too many areas e.g., wellbeing. Or inequalities opportunities 

Thank you for considering the focus of our manuscript. Similar to our response to point 1.3, we feel the manuscript provides a true reflection of the wide ranging impacts of school closures on children. Therefore, we would like to maintain this broad focus of the impacts across the domains of child development, health and education outcomes and think this fits well within the inter-disciplinary scope of PLOS ONE. 

1.13 Need to explain the relevance of FSM for international audience 

Thank you for drawing this to our attention and we agree further detail about FSM and it’s relevance is important for international readers. To address your comment, we have provided additional information in the ‘Participants’ section, outlining how eligibility criteria is determined in Wales and a reference to a Government website. In addition, we have also referenced that it is widely considered a proxy for socio-economic disadvantage [29]. Finally, we have stated that we manually obtained FSM information by inputting the school name provided into a Government website ‘My Local School’, purpose of providing contextual information about the sample. 

• “84% of the sample provided school name. This was used to manually obtain information regarding the proportion of pupils eligible for free school meals (FSM) using the ‘My Local School’ government website[28], and ranged from 1.7-58.7% (national average 19%). Eligibility criteria for FSM in Wales is based upon households receiving state benefits such as Income Support and FSM is commonly used as proxy measure of socio-economic disadvantage [29,30].” (Page 7, line 204-211)

1.14 Don’t include too much ethics detail –sought and approved is enough

Thank you for offering this suggestion. We feel it is important to retain the ethical information regarding participation, consent, ethical approval and issues relating to personal data. 

1.15 At beginning of discussion re-state key literature this was missing

We appreciate you raising this to our attention and agree that discussing some of the key literature is important to include at the beginning of the discussion. To address your comment, we have re-stated information regarding the global scale of school closures and outlined evidence that children appear to have lower susceptibility to infection. We have also highlighted the wide range of negative impacts of school closures across the domains of physical and mental health, educational attainment and economic outcomes. Finally, we have included references to UK, European and USA data about widening educational inequalities, and the impact on future earnings. 

• “The COVID-19 pandemic resulted in the implementation of national school closures in more than 190 countries and impacting over 1.6 billion pupils globally[1]. Limited evidence exists to support school closures [19] as children under the age of 14 appear to have lower susceptibility to infection [2–6]. Closing schools also contributes towards widening inequalities and inequities, and negatively impacts children’s physical and mental health, educational attainment, and social and economic outcomes [8–10,14,15]. Globally, widening inequalities in education have been reported in the UK [15], across Europe [16,17] and the USA[18], and it is estimated that school closures result in an average loss of between 0.3 and 0.9 quality-adjusted years of schooling and a reduction in potential earnings during adulthood[11]. 

The reopening of schools requires research to improve understanding of the impact of school closures on the education, health and wellbeing of pupils and school staff. Furthermore, this needs to acknowledge the challenges posed during reopening and how countries can return to a “new (ab)normal”, informing emerging policy and practice [38]. This study proposes five recommendations based on the perspectives, experiences and suggestions of primary school staff during the period of school closures (March to June 2020) and the initial phased reopening of schools in the summer term 2020. These recommendations aim to identify priorities to inform policy and practice in post-COVID education provision." (Page 24, line 850-878)

1.16 Put survey questions in appendix not in text

Thank you for highlighting this. We have addressed your comment by including a copy of the full survey in Supplementary File 1. 

• Supplementary File 1: School staff survey

Reviewer #2

2.1 Title: the use of the terms ‘national qualitative’ may be a bit of a conflict as by calling it ‘national’ implies representativeness, which qualitative work does not seek to do. It may be more helpful to specify that it was conducted in Wales within the title.

Thank you for raising this point. We agree with your suggestion and have amended as such to the following;

• “Primary school staff perspectives of school closures due to COVID-19, experiences of schools reopening and recommendations for the future: a qualitative survey in Wales.”

2.2 Introduction: The first paragraph was extremely long and it was difficult to follow the line of argument – could they break this paragraph up a bit?

We agree that the first paragraph needed to be broken up into smaller paragraphs to improve its readability. We have addressed your comment and divided this into a number of paragraphs on page 1 and 2. 

2.3 Introduction: Could the authors explain what the “continuity of learning” policy was and what it involved in practice for school staff and families?

This is an important point you have raised as we want to ensure that all terms used are understandable to a range of audiences. We have incorporated your suggestion by including a short description of the policy. 

• “This “Stay Safe. Stay Learning” strategy focused on the safety, physical and mental health and wellbeing of learners and the wider education workforce, as well as providing the principles for maintaining learning, and the transition back to schools at the appropriate time [23].” (Page 3, line 79-82)

2.4 Introduction: The tense changed from past to current

Thank you for drawing this to our attention. We have adjusted the tense where appropriate, and distinguished between past events (e.g. school closures) and current evidence. For example, we have removed ‘has’ from the sentence below; 

• “The COVID-19 global pandemic caused by the severe acute respiratory syndrome coronavirus 2 (SARS-CoV-2) resulted in the temporary closure of education across all settings and a rapid move to “emergency remote teaching”” (Page 2, lines 28-29)

2.5 Method: The survey was distributed in July but teachers were described as reporting on the re-opening of schools in September (line 89) – could the authors provide dates of when the survey was completed?

Thank for raising this, we agree that this required further clarity and we addressed your comment by including dates regarding when the survey was completed, and clarified that experiences were gathered regarding the initial phased reopening of schools in June to July 2020. 

• “The survey was formulated to explore perceptions of the impact of school closures (between March and June 2020) and gather early experiences of primary school staff during the phased reopening of schools (June to July 2020) in order to develop recommendations regarding the full-time return in September 2020.” (page 4, line 126-145) – and restated with each description of the research aim

• “The survey was open for responses between 9th and 31st July 2020. Inclusion criteria for participating in the survey was any primary school staff member (headteachers, deputy headteachers, teachers, teaching assistants, support staff) working within a local authority maintained primary school in Wales, UK.” (Page 5, line 148-151)

2.6 What were the inclusion/exclusion criteria for taking part?

This is an important point to include and we have addressed your suggestion by providing detail regarding inclusion/exclusion criteria for survey participation.

• “Inclusion criteria for participating in the survey was any primary school staff member (headteachers, deputy headteachers, teachers, teaching assistants, support staff) working within a local authority maintained primary school in Wales, UK.” (Page 5, line 148-151)

2.7 Method: Although the sampling method was described as purposive, it appeared that it was based more on convenience sampling (i.e., all schools in Wales were contacted). To what extent did they manage to capture important perspectives from difficult to reach people?

Thank you for raising this. Whilst we perceived that our sampling method was purposive in terms of seeking primary school staff participation by contacting all primary schools, we agree that our findings are based on a convenience sample of those that chose to participate. We have corrected this within the manuscript, and included this within the ‘Strengths and limitations’ section. 

• “A convenience sample of staff (headteachers, deputy headteachers, teachers, teaching assistants, support staff) working in primary schools across Wales were invited to participate in the HAPPEN School Staff survey in July 2020.” (Page 6, line 183-185)

• “Although all schools in Wales were contacted with details regarding the staff survey, the findings in this study only represent a convenience sample of those that participated.” (Page 35, line 1265-1266)

You also draw attention to an interesting point regarding methods to engage with wider groups and capture perspectives from those harder to reach. We used a number of recruitment methods in order to increase the scope of the survey reach, and we have outlined the three stage approach used for disseminating the survey. For example, we aimed to maximise our collaborations with key stakeholders through the HAPPEN primary school network, sharing the survey through their networks and existing partnerships with schools. This remains a challenge in all school-based research, and we aim to further extend collaborations through the HAPPEN network for future research projects. 

• “A combination of recruitment methods were used in order to increase the reach and uptake of the survey aside from pre-existing schools engaged with HAPPEN. Initially, all primary schools in Wales (which include pupils from ages 3-11) were contacted via email through HAPPEN inviting their staff to complete the survey. A snowball method was encouraged in which the representative from each school was asked to share the survey with staff within the primary school. Next, the survey was shared with key education stakeholders (e.g. local authorities, regional education consortia) to disseminate through their networks and existing partnerships. Finally, a social media campaign was delivered using paid advertisement on Facebook and Twitter.” (Page 6, line 185-193)

2.8 Methods: Attempts to recruit participants and reasons for non-participation should be stated.

Similar to our response to point 2.7 regarding our recruitment strategy, we used a three step process which we have amended and described in more detail in the manuscript. Firstly, all primary schools in Wales were contacted via email, secondly the survey was shared through existing stakeholder partnerships (outlined above) and finally, we developed a social media campaign using paid advertisement. We did not capture reasons for non-participation in this study, and have amended the manuscript to state this important point.

• “A combination of recruitment methods were used in order to increase the reach and uptake of the survey aside from pre-existing schools engaged with HAPPEN.” (Page 6, line 185-187)

• “Reasons for non-participation were not captured at a school or individual staff level.” (Page 7, line 197-198)

2.9 Methods: Second coding was described as being for ‘accuracy’; however, qualitative research wouldn’t typically expect ‘accuracy’ and instead would recognise there is no one correct way to interpret data.

We acknowledge the wording used to describe coding is not the aim of qualitative research and have removed this from the manuscript. We have amended this to explain how the two researcher coding interpretations were compared to account for consistencies and inconsistencies. 

• “These coding responses were compared between researchers to account for consistencies or inconsistencies in their interpretations of responses and associated codes.” (Page 9, line 291-293)

2.10 Methods: The theoretical framework for the analysis was not clearly explained and there were no references related to the methodology

We agree that the manuscript requires a theoretical description of our approach to coding. We have clarified this by outlining and describing the six stage process proposed by Braun and Clarke that was followed, from initial re-reading of responses for immersion in the data, to the generation of initial codes and ideas that formulated the final recommendations. We have also included a number of references to methodological literature including Braun and Clarke (2006) and Nowell et al (2017). [32-36].

• “This research study adopted an interpretive approach through thematic analysis of open-ended survey responses. Thematic analysis is considered a suitable approach in gaining meaningful comprehension of patterns in participant perspectives captured using a range of qualitative methods including open-ended survey questions[31,32]. Thus, using this approach this study aimed to elicit participants’ experiences relating to school closures, the initial phased reopening of schools and their recommendations to inform the full-time return to education.

The lead (EM, PhD) and second (CT, MSc) researcher involved in data analysis were both female and had previous experience in conducting thematic analysis in qualitative school-based research. The researchers did not have any interaction with participants. Responses from the school staff survey were downloaded to an Excel spreadsheet. Participants were assigned a unique study ID and identifiable information removed. The coding process followed the six steps outlined by Braun and Clarke [32]. Firstly, the lead researcher (EM) systematically read responses several times to facilitate immersion in the data. Re-reading responses allowed familiarity with the data, and the use of memoing recorded notes of patterns and emerging insights relating to coding ideas. Thoughts relating to decision processes were documented in a reflexive journal [33,34]. These documented notes formed the basis of the second stage of coding, whereby initial codes using words and phrases were manually assigned to participants’ responses. The second researcher (CT) independently coded random survey responses from 50 participants. These coding responses were compared between researchers to account for consistencies or inconsistencies in their interpretations of responses and associated codes. Agreement between researchers was very high; if there was a discrepancy or disagreement, a third researcher adjudicated (SB). This method attempts to reduce researcher bias and enhances the validity of codes and categories assigned[35]. Following this, the third stage of coding involved the two researchers working together through an extensive process of collating, combining and categorising codes into over-arching themes and sub-theme headings. In cases where codes were merged, removed or renamed, notes were taken in a reflexive journal and raw files were saved as new versions and archived with dates in order to keep an audit trail of the coding process[34]. Throughout these processes, a codebook and manual was developed to record codes and themes. The themes and sub-themes were further refined in stage four to ensure clear distinctions between themes and then grouped into over-arching recommendations based on school staff suggestions for future school closures and the reopening of schools. The lead researcher then manually reviewed the open-ended survey responses and coded according to the final list of recommendation themes and sub-themes. These responses containing themes and sub-themes were compiled to a master copy document that was used as a reference for stages 5 and 6 whereby themes were written and discussed within the findings. 

To ensure trustworthiness, the grouping of survey responses into recommendation themes developed by the researchers were screened by a primary school. The interpretation and discussion of themes and the final manuscript was also peer-reviewed for member checking by a primary school headteacher (study participant, RD) for corrections, clarification or confirmation [34,36].” (Page 8, line 231-313) 

2.11 Methods: It would be good to justify the use of this analysis for survey response data

Thank you for raising this important point, we agree that offering a justification for the choice of analysis for this data is essential. To address this, we have included additional justification within the first paragraph of the ‘Data analysis’ section and additional references [31-32]. For example, thematic analysis is considered a suitable approach in gaining meaningful comprehension of patterns in participant perspectives captured using qualitative methods including open-ended survey questions. 

• “This research study adopted an interpretive approach through thematic analysis of open-ended survey responses. Thematic analysis is considered a suitable approach in gaining meaningful comprehension of patterns in participant perspectives captured using a range of qualitative methods including open-ended survey questions[31,32]. Thus, using this approach this study aimed to elicit participants’ experiences relating to school closures, the initial phased reopening of schools and their recommendations to inform the full-time return to education.” (Page 8, line 231-237)

2.12 Methods: Given that qualitative researchers closely engage with the research process, it would be helpful to know more about the research team and provide evidence of reflexivity in the process of analysis.

Given that the researcher is the instrument for analysis in qualitative research, we agree that offering additional disclosure about the processes undertaken to ensure reflexivity is paramount. We have outlined further information that will be useful for the reader to consider in interpreting reflexivity. These include information about the researchers responsible for coding, maintaining a reflexive journal to record thought processes relating to coding and keeping an audit trail of the data analysis process. 

• “The lead (EM, PhD) and second (CT, MSc) researcher involved in data analysis were both female and had previous experience in conducting thematic analysis in qualitative school-based research. The researchers did not have any interaction with participants.” (Page 8, line 238-240)

• “Firstly, the lead researcher (EM) systematically read responses several times to facilitate immersion in the data. Re-reading responses allowed familiarity with the data, and the use of memoing recorded notes of patterns and emerging insights relating to coding ideas. Thoughts relating to decision processes were documented in a reflexive journal [33,34].” (Page 8, line 243-247)

• “In cases where codes were merged, removed or renamed, notes were taken in a reflexive journal and raw files were saved as new versions and archived with dates in order to keep an audit trail of the coding process[34]. Throughout these processes, a codebook and manual was developed to record codes and themes.” (Page 298-300)

2.13 Methods: Did researcher use memo-ing in order to demonstrated how they developed their understanding of the data?

Thank you for raising this and we agree we did not provide sufficient detail relating to data coding processes. We have addressed this comment which can be found on the page listed below. 

• “Re-reading responses allowed familiarity with the data, and the use of memoing recorded notes of patterns and emerging insights relating to coding ideas.” (Page 8, line 245-246)

2.14 Methods: Did the team obtain feedback from participants on the research findings to validity to the team’s interpretations?

This is an important point and something that we incorporated into our data analysis and interpretation of findings, listed below. Furthermore, we invited a primary school headteacher as a co-author on the manuscript (RD), who provided critical input to the overall interpretation and discussion of findings from a practice perspective during the co-author peer review process. 

“To ensure trustworthiness, the grouping of survey responses into recommendation themes developed by the researchers were screened by a primary school. The interpretation and discussion of themes and the final manuscript was also peer-reviewed for member checking by a primary school headteacher (study participant, RD) for corrections, clarification or confirmation[34,36]”. (Page 9, line 309-313)

2.15 Results: The results were framed as recommendations for the future so I wonder whether the research questions (that related to experience of school closures as well as recommendations) need to be a bit clear that the focus is on using the experiences to generate recommendations.

This is very useful feedback as we want to ensure that the purpose of our study and research question are clear to the reader. We have addressed this comment by rewording the research question and other instances where mentioned; 

• “The aim of this qualitative study is to reflect on the perspectives of primary school staff regarding school closures between March and June 2020 and their experiences of the initial phased reopening of schools between June and July 2020 to identify recommendations for the full-time reopening of schools from September 2020 or future scenarios of school closures and blended learning.” (Page 4, line 117-121)

• “The survey was formulated to explore perceptions of the impact of school closures (between March and June 2020) and gather early experiences of primary school staff during the phased reopening of schools (June to July 2020) in order to develop recommendations regarding the full-time return in September 2020.” (Page 4, line 126-145) 

• “Thematic analysis was used to generate themes of recommendations based on survey responses to gain an understanding into the perspectives and experiences of primary school staff during the period of school closures (March to July 2020) and the phased reopening of schools (June to July 2020) to inform the full-time return to education in September 2020.” (Page 5, line 151-155)

• “This study proposes five recommendations based on the perspectives, experiences and suggestions of primary school staff during the period of school closures (March to June 2020) and the initial phased reopening of schools in the summer term 2020.” (Page 25, line 874-878)

2.16 Results: With some of the quotes, it wasn’t clear to what extent they reflected staffs’ experiences versus their expectations, e.g, “the children will have been affected by being out of school for

Thank you for raising this point and we agree that some quotes were ambiguous and require specific wording to ensure that this is clear. We have addressed this throughout the findings section, with examples outlined below. 

• “Many teachers perceived that pupils were engaging in less physical activity during lockdown” (Page 10, line 336-337)

• “In addition, weight gain was observed by a number of teachers upon the phased return to school” (Page 10, line 338-339)

• “Staff also perceived increased screen time” (Page 10, line 340-341)

• “Others outlined their expectations for the full-time return to school;” (Page 11, line 361-362)

• “Whilst staff highlighted concerns about learning regression, discussed in Recommendation 2, prioritising health and wellbeing was still advocated for upon the return to school, particularly with the uncertainty of the longer-term impacts of COVID-19 on the wider education system and expectations by staff” (Page 12, line 393-396) 

• “School staff highlighted challenges they had experienced when combining work and home life” (Page 13, line 454-455)

• “Staff felt that scientific evidence would make the full-time return to school easier;” (page 14, line 499-500)

• “Experiences of high parental engagement were also reflected in improved relationships with and communication between the school and parents” (Page 15, line 546-547)

• “However, school staff reported wide variations in home learning provision;” (Page 15, line 555)

• “Staff experiences during remote teaching and learning included” (Page 18, line 662)

• “School staff experiences during the phased reopening of schools were captured, including adaptations to regular teaching practice (e.g. smaller class sizes) and the learning environment (e.g. higher staff to pupil ratio) that benefitted pupils.” (Page 20, line 713-715)

• “Finally, concerns relating to expectations about the enhanced cleaning and hygiene practices required in the full-time return to school were also remarked upon” (Page 22, line 777-778)

2.17 Results: 186 so long, and will have trouble re-adjusting to school life” (line 186). I would be cautious about interpreting this as young people having had trouble adjusting – but more than staff expected this to be the case.

We appreciate you drawing this to our attention. We have re-worded this to reflect staff expectations as the following; 

• Others outlined their expectations for the full-time return to school; “the children will have been affected by being out of school for so long, and will have trouble re-adjusting to school life” (year 2 teacher).” (Page 11, line 361-362)

2.18 Results: There were other quotes where it wasn’t always clear that this reflected the theme, e.g., “a lot of time spent on computers and not going outside” (year 2 teaching assistant) (line 199) may just reflect lockdown and may not mean the child requires anything other than coming out of lockdown.

Thank you for providing this insight. We feel this quote fits within the umbrella of physically active/sedentary behaviours. We have moved this to be outlined following the reporting of physical activity and weight gain. 

This is one of the challenges with the health and wellbeing recommendation theme. Whilst staff perceive children were engaging in unhealthy behaviours, this doesn’t necessarily mean children will maintain these behaviours. A longer term follow up including objective measurements of these behaviours would be an important avenue for future research in order to understand whether these impacts were temporary e.g. reflective of a change in routine, or long term. 

However, the findings reflect the importance of the school environment in create structure, stability, routine, opportunities for being active, play etc that children were stripped of during school closures.

• “Many teachers perceived that pupils were engaging in less physical activity during lockdown, with responses to the negative effects of school closures including “less active” (year 1 teacher), and a “lack of physical fitness” (reception teacher). In addition, weight gain was observed by a number of teachers upon the phased return to school; “some children have put on an unhealthy amount of weight” (year 2 teaching assistant). Staff also perceived increased screen time and gaming behaviours with responses including “a lot of time spent on computers and not going outside” (year 2 teaching assistant), and “Playing inappropriate games and excessive screen time” (year 3 teacher).” (Page 10, line 336-354)

2.19 Results: It would be helpful to better understand how the quotes about gaps in learning (p216) fitted with the recommendation around a focus on wellbeing – rather than a focus on maintaining/improving learning during lockdown

Thank you for pointing this out as we have reconsidered that discussions about gaps in learning are better placed within the recommendation relating to parental support and engagement. This was already discussed within the parental engagement findings in the discussion in the original submission. To address your point we have moved these discussions to Recommendation 2: 

• “However, school staff reported wide variations in home learning provision; “For those children who have had parents who have supported them with online learning this has been great”. Discussions on this topic included the inequalities and inequities present and a sense of widening gaps in learning was projected; “Gaps in learning depending on parental support” (deputy headteacher). Another headteacher affirmed that for these pupils, “their home life has provided limited stimulation”. A significant concern was the regression of key skills and knowledge during the period of school closures based on staff observations during remote teaching; “Many not accessing learning and falling behind or regressing” (reception support staff), and “their motivation has reduced. Key skills deteriorated. Knowledge lost” (year 3 teacher). As a result, concerns for the full-time return to school included; “children's wellbeing, gaps in learning - especially between groups of learners who have engaged and those who haven't with home learning” (year 4 teacher). Increasing inequalities were also highlighted regarding children with ALN as “some have had no learning as have either not the support at home to help or have refused to work outside of the school setting…list is endless working at a SEN [special educational needs] school” (teaching assistant) and “The vulnerable and ALN pupils generally have not engaged in the sessions” (headteacher).” (Page 15, line 555-588)

2.20 Results: Results were very much at the summary level, but given the purposive sampling, I was interested to know whether there was variability in experiences/perspectives on the basis of the sampling characteristics

We have amended our sampling method (outlined in point 2.7) as although all primary schools in Wales were contacted, findings reflect those that chose to participate and therefore we have reconsidered this as convenience sampling. 

We appreciate this observation as it has drawn to our attention that further clarity is required regarding the school characteristics of free school meal (FSM) eligibility. Details regarding FSM were not collected as part of the research, but rather obtained from a Government website (My Local School) using school name provided by the participants, to provide contextual information about the sample. 

We acknowledge that the previous wording e.g. “Information regarding the proportion of pupils eligible for free school meals (FSM) within each school was obtained from….” leads the reader to consider this as part of the sampling strategy. We have amended this to make it clear that this information was obtained through ‘My Local School’ government website following participation in the survey. 

• “84% of the sample provided school name. This was used to manually obtain information regarding the proportion of pupils eligible for free school meals (FSM) using the ‘My Local School’ government website[28], and ranged from 1.7-58.7% (national average 19%). Eligibility criteria for FSM in Wales is based upon households receiving state benefits such as Income Support and FSM is commonly used as proxy measure of socio-economic disadvantage [29,30]. 50% of school staff worked in schools with a proportion of pupils eligible for FSM greater than the national average (>19%).” (Page 7, line 204-211)

Therefore, variations in experiences/perspectives is not within the scope of this research study or the research question, but we agree that this would be a very interesting topic for future research. 

2.21 Discussion: Whilst it would be ideal for schools to receive advance notice around changes in guidance, it may be helpful to recognise the ever-changing nature of the pandemic – so although ideal to ‘ensure schools receive advance notice of changes in guidance’ this may not always be possible.

We agree with this – whilst it is an important point and was advocated for by school staff, it is also important to consider the practicalities with schools receiving advance notice. We have addressed this comment to discuss some of the challenges with this such as the rapidly changing nature of the pandemic and the reactive response to rapidly emerging data. 

• “School staff advocated for receiving regular and clear guidance and advance notice of changes in guidance to allow preparation time, to facilitate communication with parents and to enable schools to manage parental expectations more effectively. Whilst it is likely that this is also relevant across different sectors impacted by changes to working and operation, it is important to acknowledge the practicalities associated with a rapidly changing pandemic and subsequent governmental decision making. Decisions regarding education delivery including those relating to school closures, reopening or adaptations to practice are often reactive to rapidly emerging data and thus, communicating changes is most efficient through national media coverage.” (Page 33, line 1207-1222) 

We have also deleted from abstract and conclusion, as although this was suggested by staff, the general recommendation of improving communication between government and schools encapsulates some of the mechanisms behind advocating for advance notice (that may not be practical in other pandemic scenarios).

Best wishes,

Emily Marchant

Corresponding author

---

## [Decision Letter · Decision Letter 1]

28 Apr 2021

PONE-D-20-34984R1

Primary school staff perspectives of school closures due to COVID-19, experiences of schools reopening and recommendations for the future: a qualitative survey in Wales.

PLOS ONE

Dear Dr. Marchant,

Thank you for submitting your manuscript to PLOS ONE. After careful consideration, we feel that it has merit but does not fully meet PLOS ONE’s publication criteria as it currently stands. Therefore, we invite you to submit a revised version of the manuscript that addresses the points raised during the review process.

In constructing your revision, I urge you to consider the comments provided by Reviewer #2, particularly with regards to the aim and significance of the study and presentation of findings as recommendations. 

We look forward to receiving your revised manuscript.

Kind regards,

Amanda A. Webster

Academic Editor

PLOS ONE

Journal Requirements:

Reviewers' comments:

Reviewer's Responses to Questions

**Comments to the Author**

1. If the authors have adequately addressed your comments raised in a previous round of review and you feel that this manuscript is now acceptable for publication, you may indicate that here to bypass the “Comments to the Author” section, enter your conflict of interest statement in the “Confidential to Editor” section, and submit your "Accept" recommendation.

Reviewer #1: (No Response)

Reviewer #2: All comments have been addressed

2. Is the manuscript technically sound, and do the data support the conclusions?

Reviewer #1: Partly

Reviewer #2: Yes

3. Has the statistical analysis been performed appropriately and rigorously? 

Reviewer #1: N/A

Reviewer #2: N/A

4. Have the authors made all data underlying the findings in their manuscript fully available?

Reviewer #1: Yes

Reviewer #2: Yes

5. Is the manuscript presented in an intelligible fashion and written in standard English?

Reviewer #1: Yes

Reviewer #2: Yes

6. Review Comments to the Author

Reviewer #1: Thank your responding to my recommendations. I do however still have a few concerns. My main concern is the lack of theoretical underpinnings --you are presenting the findings as empirical data yet your are seeking to reflect 'perspectives' of the teachers in a school. You slipped in the idea of interpretative approach to your study. My suggestion here is that you need to unpack interpretivism as your theoretical framing --both in regard to how you did your coding/analysis but also in how it was used to inform/shape the inferences drawn in your discussion. You need to tell the reader why this approach was most appropriate to your study and how it was used. This could go around line 81. After this you can add your section on aim of your study. Finish with your research question. In the section on the aim I would like to see an explicit stated link back to your introduction --e.g., why was it necessary to reflect on the perspectives of primary school staff - simply to make recommendation is not enough --this is where you talk about our gap in understanding the conditions and implications of shut down and COVID on schools --this needs to be make clear.

I would also like to see you change background to introduction.

In the results I still feel as if I am getting lost in the quotes - I suggest you start with either a figure/diagram/table or summary para --where you outline the key findings overall.

I would remove the word 'recommendation' from the key finding headings in the finding section and leave this term/terminology to the discussion. Otherwise it starts to sound like a report and not an academic paper.

I would remove dot points from conclusion - write as a narrative.

I am not familiar with the references style used? the use of dot points and numbers in the references list is this correct??

I do believe you have a good study with some important information ---the use of theoretical underpinning and a tightening up in the presentation of the findings for clarity would enhance this paper.

All the best

Reviewer #2: Thank you for a further opportunity to review the manuscript. I think the authors have clearly and thoughtfully addressed my comments and am happy with their revisions.

7. PLOS authors have the option to publish the peer review history of their article (what does this mean?). If published, this will include your full peer review and any attached files.

Reviewer #1: **Yes: **Lynn Sheridan

Reviewer #2: **Yes: **Polly Waite

---

## [Author Response · Author response to Decision Letter 1]

3 Jun 2021

Dear Dr Webster,

On behalf of all authors, we would like to thank both reviewers for taking their time to read our revised manuscript. Reviewer #2 was happy that we addressed their comments. We would like to extend our appreciation to Reviewer #1 who provided detailed feedback and comments in a second round of review of our manuscript. We feel this added input has strengthened the manuscript, particularly in relation to providing detailed information about the study’s theoretical underpinning, and improving the clarity of the findings with an additional summary, summarising key findings at the start of each sub-section. Please find below our response to reviewer #1, outlining changes made and the associated page/line numbers within the tracked changes submission for cross-reference. Furthermore, we have made some minor amendments to wording and included some relevant new literature that has been published since the original submission (e.g. [82] Sundaram et al 2021, page 37, line 920; and [65] Watermeyer et al 2021, page 31, line 776). 

1.1 Reviewer #1: Thank your responding to my recommendations. I do however still have a few concerns. My main concern is the lack of theoretical underpinnings --you are presenting the findings as empirical data yet your are seeking to reflect 'perspectives' of the teachers in a school. You slipped in the idea of interpretative approach to your study. My suggestion here is that you need to unpack interpretivism as your theoretical framing --both in regard to how you did your coding/analysis but also in how it was used to inform/shape the inferences drawn in your discussion. You need to tell the reader why this approach was most appropriate to your study and how it was used. This could go around line 81. After this you can add your section on aim of your study. Finish with your research question.

Thank you for raising this important point regarding the theoretical underpinning of this study. We agree that offering a rich description of our theoretical approach would help the reader to interpret the findings with a greater degree of understanding. To address your suggestion of unpicking the interpretative approach, we have included additional theoretical information, including the use of a grounded theory approach in order to develop the recommendations in which our findings are based on. We now feel that we have justified our theoretical standpoint in relation to the framing of the study, coding, analysis and in informing the inferences from our study. Please find below specific revisions and page numbers to address your comment. 

Introduction

• Grounded theory is a useful approach when little is known about a phenomenon[27]. In this case of this study, the rapidly evolving impacts of the COVID-19 pandemic brought with it unprecedented changes to society including national school closures based on limited evidence of potential challenges facing schools and learners. This study uses an interpretive approach with tenets of grounded theory methodology in order to develop a pattern of meanings, explanations and descriptions that reflect the perspectives and experiences of primary school staff within the context of school closures and phased reopening of schools. The purpose of the theoretical approach used in this study is to explore and extract the different perspectives in order to develop a rigorous and robust foundation for shaping and influencing emerging policy and practice in post-COVID education provision. Thus, the pragmatic approach to grounded theory in assessing and addressing challenges in practice [28] is suitably aligned to the aims of this study.

The aim of this grounded theory qualitative study is therefore to reflect on the perspectives of primary school staff regarding school closures between March and June 2020 and their experiences of the initial phased reopening of schools between June and July 2020 to identify recommendations for the full-time reopening of schools from September 2020 or future scenarios of school closures and blended learning (page 4, line 91-117). 

Methods

• The basis of this study relies on an interpretivist approach, whereby the authors assume social reality is shaped by human experiences (primary school staff) and social contexts (primary school closures). Grounded theory research gathers data relating to participants’ different lived experiences in order to generate plausible theory to understand the contextual reality of common circumstances[26]. A grounded theory approach was applied to the data collection, analysis and explanatory findings in order to offer descriptions of the perspectives and experiences of primary school staff during the period of national school closures (between March and June 2020) and the phased reopening of schools (June to July 2020). 

Generating theory in grounded theory relates to plausible relationships proposed amongst concepts and sets of concepts[29]. In the case of the current study, the generation of theory relates to the development of recommendations regarding the full-time return from September 2020, based on these lived experiences of primary school staff. Therefore, this approach intends to inductively construct these recommendations aligned to these lived experiences to contribute to the limited evidence base of the impact of school closures and to shape and influence emerging policy and practice during the ‘education recovery’ period following the COVID-19 pandemic. (page 5, line 122-139) 

Methods – School staff survey

• An online survey with primary school staff was distributed in July 2020 through the HAPPEN primary school network (Health and Attainment of Pupils in a Primary Education Network) [30]. The survey was open for responses between 9th and 31st July 2020. Inclusion criteria for participating in the survey was any primary school staff member (headteachers, deputy headteachers, teachers, teaching assistants, support staff) working within a local authority maintained primary school in Wales, UK. A grounded theory strategy using thematic analysis was used to generate themes of recommendations based on survey responses to gain an understanding into the perspectives and experiences of primary school staff during the period of school closures (March to July 2020) and the phased reopening of schools (June to July 2020) to inform the full-time return to education from September 2020. (page 6, line 141-150)

• Given that interpretivist grounded theory is based on inductively developing patterns of meaning, a range of broad, open-ended questions were included to explore the research question. As grounded theory presumes limited preconceived ideas of the research topic, the survey incorporated polarised open-ended questions regarding school closures (page 6, line 184-187)

Methods – Data analysis

• This research study adopted an interpretivist approach through thematic analysis of open-ended survey responses. Thematic analysis is considered a suitable approach in gaining meaningful comprehension of patterns in participant perspectives captured using a range of qualitative methods including open-ended survey questions [35,36]. Thus, this study used thematic analysis as a data analysis technique to elicit participants’ perspectives and experiences relating to school closures and the initial phased reopening of schools. In providing context-specific explanations and descriptions of the impacts of school closures, this study used an overarching grounded theory methodology to inductively develop theory presented as recommendations to inform the full-time return to education and emerging policy and practice. Thematic analysis as an analytical approach in grounded theory studies has been used widely in other settings-based research [28]. (page 9, line 239-255)

• This stage incorporated a grounded theory approach whereby both researchers revisited the coded survey responses and followed a constant comparative method to progressively categorise and refine the themes and sub-themes. (page 10, line 275-279)

1.2 In the section on the aim I would like to see an explicit stated link back to your introduction --e.g., why was it necessary to reflect on the perspectives of primary school staff - simply to make recommendation is not enough --this is where you talk about our gap in understanding the conditions and implications of shut down and COVID on schools --this needs to be make clear.

Thank you for this suggestion, we agree that an explicit stated link between the aim of our research study in addressing a gap in understanding of the implications of school closures would be clearer for the reader. Please find details below of this revision:

• The implementation of national school closures as a public health measure to reduce social contacts resulted in a range of unintended consequences. Evidence demonstrates the impact on children’s development, physical and mental health, social and economic outcomes and widening inequalities [8–12,14,15]. However, there is limited evidence regarding the challenges posed to schools during school closures and reopening, and a gap in understanding of the strategies for schools returning based on these impacts [22]. This study therefore aimed to address this gap in evidence. (page 4, line 91-97)

1.3 I would also like to see you change background to introduction.

Thank you, we have amended this. 

1.4 In the results I still feel as if I am getting lost in the quotes - I suggest you start with either a figure/diagram/table or summary para --where you outline the key findings overall.

We have addressed this useful point by incorporating a summary paragraph at the beginning of each findings sub-heading:

Prioritise the health and wellbeing of pupils and staff

• A key priority highlighted by school staff was the importance of prioritising the health and wellbeing of pupils and staff. In summary, school staff consistently mentioned a range of negative impact of school closures on children. This included physical health concerns such as decreased physical activity and fitness, weight gain and lethargy, mental health concerns including increased anxiety and lower wellbeing, and the impact on social development such as a lack of social interaction, excessive gaming and social media use. Additional challenges were also highlighted regarding children with additional learning needs (ALN). The impact on school staff wellbeing included challenges with work/life balance such as the pressure of providing online teaching and learning whilst balancing personal issues and family life. Therefore, the recommendation to prioritise the health and wellbeing of pupils and staff included suggestions of specific wellbeing activities for pupils, ensuring there are designated staff responsible for wellbeing, protecting staff breaks and increased provision of outdoor learning and play. This was also emphasised in relation to staff perceptions that the return of face-to-face education would prioritise the rhetoric of “catching up” through intensive teaching and learning at the detriment of whole-school health and wellbeing. This will be discussed in more detail below. (page 11, line 297-320)

Focus on enabling parental support and engagement

• The importance of the home learning environment during school closures was highlighted as a key factor in children’s learning progression. This is outlined in the findings from this recommendation theme in which primary school staff noted widening inequities and inequalities. Staff in this study commented that the large variation in home learning environments and parental support/engagement accounted for gaps in children’s learning. This theme encompasses aspects of parental support and engagement that are both positive; one-to-one support, wider skill development, improved parent/school relationships, parental awareness of children’s learning needs, and negative; key skill regression, lack of engagement with/decreased motivation for learning. School staff acknowledged a wide range of barriers for parents and families supporting home learning, including working commitments and language barriers. Therefore, staff suggested introducing support systems such as advice helplines for parents, and emphasised the importance of strengthening links between schools and families. (page 16, line 425-438)

Enhance (pupil, parent and staff) digital competence through increased access and support

• During lockdown and school closures, teaching was primarily delivered through digital methods requiring children to engage in remote learning online. Also associated with children’s learning progression and gaps in learning, this recommendation theme covers a range of digital barriers. Pupils experienced digital exclusion through a lack of access to online learning materials and digital equipment (e.g. internet access, laptop) and competing demands for equipment in the household. Staff in this study also highlighted that digital competency was an issue for pupils, teachers and parents, with a lack of digital understanding hindering pupils’ ability to learn, teachers’ confidence and ability to teach online and parents’ ability to support their child with online home learning. Whilst staff suggested the provision of digital equipment and internet access, they emphasised that this should be matched with digital training for all, including teacher training for blended learning and training to support children and parents accessing online learning materials. This will be discussed in more detail below. (page 20, line 514-527)

Adapt the learning environment and teaching practice

• This recommendation theme represents school staff experiences during the phased reopening of schools, including adaptations to regular teaching practice (e.g. smaller class sizes) and the learning environment (e.g. higher staff to pupil ratio) that benefitted pupils. Staff in this study suggested that decreasing class sizes and increasing staff numbers would support whole-school wellbeing and allow learning support targeted to need. However, staff acknowledged that this required significant government investment and support to facilitate longer term adaptations to education provision. (page 23, line 578-584)

Clear communication of guidance and expectations

• This final theme constitutes the importance of clear and regular communication between government and schools, the possibility of schools receiving advance notice of changes in guidance to support planning and expectations, and the challenges of incorporating generic guidance to fit within the contextual school differences and needs. (page 25, line 636-639)

1.5 I would remove the word 'recommendation' from the key finding headings in the finding section and leave this term/terminology to the discussion. Otherwise it starts to sound like a report and not an academic paper.

We have amended this suggestion and agree that it is in line with the presentation of an academic paper.

1.6 I would remove dot points from conclusion - write as a narrative.

Thank you for this suggestion. We have addressed this by presenting the conclusion as a narrative discussion of the recommendations, removing dot points. (page 39)

1.7 I am not familiar with the references style used? the use of dot points and numbers in the references list is this correct??

This paper uses the Vancouver referencing style, in line with the requirements by PLOS. This uses dot points in-text and numbers in the reference list: 

PLOS uses the reference style outlined by the International Committee of Medical Journal Editors (ICMJE), also referred to as the “Vancouver” style. Example formats are listed below. Additional examples are in the ICMJE sample references.

https://journals.plos.org/plosone/s/submission-guidelines#loc-references

1.8 I do believe you have a good study with some important information ---the use of theoretical underpinning and a tightening up in the presentation of the findings for clarity would enhance this paper.

Thank you for this positive comment. We appreciate the time you have committed to offering a second round of revisions for our paper. We feel we have addressed your concerns, in particular regarding the theoretical underpinning of our study and believe this has significantly strengthened the manuscript. We have also provided a summary paragraph at the start of each of the findings sub-headings to ensure clarity for the reader. We also believe that the use of in-text narrative quotes will be clearer for the reader when the manuscript is typeset. 

2.1 Reviewer #2: Thank you for a further opportunity to review the manuscript. I think the authors have clearly and thoughtfully addressed my comments and am happy with their revisions. 

Thank you. 

Thank you again for the opportunity to submit a revised version of our manuscript addressing reviewers comments. We look forward to hearing from you.

Best wishes,

Dr Emily Marchant

---

## [Decision Letter · Decision Letter 2]

21 Sep 2021

PONE-D-20-34984R2

Primary school staff perspectives of school closures due to COVID-19, experiences of schools reopening and recommendations for the future: a qualitative survey in Wales.

PLOS ONE

Dear Dr. Marchant,

Thank you for submitting your manuscript to PLOS ONE. After careful consideration, we feel that it has merit and that you have addressed many of the previous concerns expressed by the reviewers, but feel that there are still a few issues that need to be resolved. Therefore, we invite you to submit a revised version of the manuscript that addresses the points raised during the review process.

I would suggest you pay particular attention to reviewer 1's comments about a research question and reviewer 3's comments about the breadth and depth of the topics examined. 

We look forward to receiving your revised manuscript.

Kind regards,

Amanda A. Webster

Academic Editor

PLOS ONE

Journal Requirements:

Reviewers' comments:

Reviewer's Responses to Questions

**Comments to the Author**

1. If the authors have adequately addressed your comments raised in a previous round of review and you feel that this manuscript is now acceptable for publication, you may indicate that here to bypass the “Comments to the Author” section, enter your conflict of interest statement in the “Confidential to Editor” section, and submit your "Accept" recommendation.

Reviewer #1: All comments have been addressed

Reviewer #3: All comments have been addressed

2. Is the manuscript technically sound, and do the data support the conclusions?

Reviewer #1: Yes

Reviewer #3: Partly

3. Has the statistical analysis been performed appropriately and rigorously? 

Reviewer #1: Yes

Reviewer #3: No

4. Have the authors made all data underlying the findings in their manuscript fully available?

Reviewer #1: Yes

Reviewer #3: Yes

5. Is the manuscript presented in an intelligible fashion and written in standard English?

Reviewer #1: Yes

Reviewer #3: Yes

6. Review Comments to the Author

Reviewer #1: Thank you so much for responding to my suggestions. I am pleased with the revisions which have improved the readability of this paper. This is important research which will offer some useful recommendations to the school system in how to operate in the COVID environment and the post COVID-'normal'.

I have one suggestion that I did make prior which I encourage you to consider and that is the inclusion of a stated central research question (i.e., line 83). As you are now using a grounded study to guide this work--you could start with: e.g.,

What were the challenges posed to schools during school closure and re-opening and how can the impact of these challenges be addressed going forward?

all the best with this important study

Lynn

Reviewer #3: I read the revised version of the paper and I can see that the authors took great care in responding addressing to the reviewer comments. As I see it, the paper has still a number of issues that remain critical. My greatest concern is that the paper reads (still) too much like a policy report and not enough like an academic paper. Of course this has to do with the fact, that authors provide a whole extensive policy recommendations (but that might me a question of taste). The second issue relates to the quality of the convenience sample. I will elaborate on these issues below.

Theoretical underpinning

A main issue in the previous round of reviews was the question of the theoretical underpinning of the study. As suggested by reviewer #1, the authors now provide further arguments for why the chosen interpretative/grounded theory approach is suitable (mostly related to the newness of Covid-19). While I agree with that premise, the fact that so many heterogenous topics are discussed (student learning, student well-being, digital learning) leads to a somewhat superficial treatment of each of them, respectively. This might be a less severe problem, if the research design would have been able to estimates of the prevalence of the different issues across as the entire population of school teachers for the selected grades - but that would have required a sufficiently large random sample of this population). In any case – I find that the manuscript still lacks theoretical depth.

The empirical foundation

I am in serious doubt to what extent the research design which is based on a convenience sample realized with snowball sampling and similar methods can really be used to “develop a rigorous and robust foundation for shaping and influencing emerging policy and practice in post-Covid education provision” – as argued by the authors. While the identified topics and different perspective identified in the paper are very interesting, no doubt, I wonder whether a different and more systematic sampling approach (either qualitative or quantitative) would have led to very different results. Or - in other words, there is a realistic chance that the teachers who participated in the study might a quite different (special) – and had a very different experience of the Covid-19 pandemic and school closures – than the rest of the teacher population. As a minimum, the authors would need to address this issue in the paper.

7. PLOS authors have the option to publish the peer review history of their article (what does this mean?). If published, this will include your full peer review and any attached files.

Reviewer #1: **Yes: **Dr Lynn Sheridan

Reviewer #3: No

---

## [Author Response · Author response to Decision Letter 2]

2 Nov 2021

Dear Dr Webster,

On behalf of all authors, we would like to again thank both reviewers for taking their time to read and comment on our revised manuscript. 

Reviewer #1:

1. Thank you so much for responding to my suggestions. I am pleased with the revisions which have improved the readability of this paper. This is important research which will offer some useful recommendations to the school system in how to operate in the COVID environment and the post COVID-'normal'.

I have one suggestion that I did make prior which I encourage you to consider and that is the inclusion of a stated central research question (i.e., line 83). As you are now using a grounded study to guide this work--you could start with: e.g. What were the challenges posed to schools during school closure and re-opening and how can the impact of these challenges be addressed going forward?

Thank you again for reviewing the second revised manuscript. We are glad that you are satisfied that we have addressed the useful comments provided in the second round of revisions. We agree that a central research question would be important and have included this in line 83 as suggested. 

Introduction. Page 4, line 86-89: This study therefore aims to address this gap in evidence by exploring what were the challenges posed to schools during school closures and reopening and how can the impact of these challenges be addressed going forward.

Reviewer #3:

2. I read the revised version of the paper and I can see that the authors took great care in responding addressing to the reviewer comments. As I see it, the paper has still a number of issues that remain critical. My greatest concern is that the paper reads (still) too much like a policy report and not enough like an academic paper. Of course this has to do with the fact, that authors provide a whole extensive policy recommendations (but that might me a question of taste). The second issue relates to the quality of the convenience sample. I will elaborate on these issues below. 

Thank you for providing feedback regarding this second revised manuscript. Our manuscript explores the central research question of what were the challenges of school closures and reopening and how can the impact of these challenges be addressed going forward. We conducted this research as a rapid qualitative study to gather the experiences of a range of primary school staff in Wales, UK during the period school closures and reopening in 2020. Our methodological approach and discussion of findings aims to address this research question most suitably in the context of the COVID-19 pandemic. 

Both the immediacy and uncertainty of the COVID-19 situation influenced our study design and theoretical underpinning, using a grounded theory approach to generate an understanding of the key challenges experienced by education practitioners in Wales as a result of national school closures. During this period and over the coming months and years, there are rapidly evolving strategies and policies developed by the government and national education bodies aiming to combat some of these challenges. These policies directly impact education practitioners in Wales, and their approaches to teaching in learning in the context of ‘education recovery’. The emergency nature of the COVID-19 pandemic and its impact on educational settings requires a research approach that contributes evidence in a manner that translates findings than can inform policy and practice that impact the education workforce. 

We acknowledge your comment that the manuscript may read in certain places as a policy report. We feel that the presentation of our study findings in the manner of a research/policy/practice paper is hugely important during the COVID-19 pandemic, a period which requires rapid reporting of evidence and data to inform emerging policy and practice. This is supported in a recent paper by Vindrola-Padros et al. (2020), ‘Carrying Out Rapid Qualitative Research During a Pandemic: Emerging Lessons From COVID-19’. The authors reflect on their experiences of conducting rapid qualitative research during COVID-19, and suggest the importance of translating and sharing evidence as “actionable findings”. The authors state that “this refers to straightforward recommendations that can be easily understood and translated into changes in policy and/or practice and requires carefully planning the use of findings during the research design phase. Even if qualitative studies are produced during epidemics, public health officials might have difficulties trusting the findings, digesting the information and translating it into changes in policy and practice” (page 2193). 

Conclusion: Page 39, Line 907-909: This work, along with related findings, has contributed to emerging policy and practice at the national scale in Wales, offering policymakers an evidence base for school-based practice into 2021 and beyond. 

To acknowledge your comment regarding the readability of the manuscript, we have included an additional explanation of why we have structured the manuscript as a set of recommendations, citing the interesting paper by Vindrola-Padros et al. (2020):

Introduction. Page: 3, Line 59-62: Furthermore, it is important for rapid research conducted during the COVID-19 pandemic to be shared as “actionable findings”, recognised by Vindrola-Padros and colleagues as “straightforward recommendations that can be easily understood and translated into changes in policy and/or practice” [22,23].

Page: 5, Line 97-99: The purpose of the theoretical approach used in this study is to explore and extract the different perspectives in order to develop a rigorous and robust foundation of “actionable findings” [24] for shaping and influencing emerging policy and practice in post-COVID education provision.

Methods. Page 6, Line 122-125: In the case of the current study, the generation of theory relates to the development of recommendations, viewed as “actionable findings” [24] regarding the face-to-face return from September 2020, based on these lived experiences of primary school staff.

Discussion. Page: 28, Line 637-638: Furthermore, this needs to acknowledge the challenges posed during reopening and develop strategies based on “actionable findings” as to how countries can return to a “new (ab)normal”, informing emerging policy and practice [24,45].

Conclusion. Page: 39, Line 904-907: It is essential for research to focus on the mechanisms that optimise and improve outcomes for children both during and after the COVID-19 pandemic, and translate these as “actionable findings” to inform emerging policy and practice [21,24].

Vindrola-Padros C, Chisnall G, Cooper S, Dowrick A, Djellouli N, Symmons SM, Martin S, Singleton G, Vanderslott S, Vera N, Johnson GA. (2020) Carrying Out Rapid Qualitative Research During a Pandemic: Emerging Lessons From COVID-19. Qual Health Res. Dec;30(14):2192-2204. doi: 10.1177/1049732320951526.

3. Theoretical underpinning

A main issue in the previous round of reviews was the question of the theoretical underpinning of the study. As suggested by reviewer #1, the authors now provide further arguments for why the chosen interpretative/grounded theory approach is suitable (mostly related to the newness of Covid-19). While I agree with that premise, the fact that so many heterogenous topics are discussed (student learning, student well-being, digital learning) leads to a somewhat superficial treatment of each of them, respectively. This might be a less severe problem, if the research design would have been able to estimates of the prevalence of the different issues across as the entire population of school teachers for the selected grades - but that would have required a sufficiently large random sample of this population). In any case – I find that the manuscript still lacks theoretical depth.

Thank you for your comments regarding the theoretical underpinning of our study. It is important to acknowledge that this manuscript has already received two rounds of revisions from two previous reviewers. During these stages of revisions, reviewer #1 raised a number of suggestions regarding additional justification and explanation of the interpretative/grounded theory approach. We are pleased that reviewer #1 is satisfied with these revisions and feels we have adequately addressed issues regarding the theoretical underpinning, including our justification of this approach particularly in the context of COVID-19. 

The use of an interpretative/grounded theory approach enabled the perspectives of education practitioners to be captured and conceptualised as key recommendations aligned to challenges and impacts of school closures and reopening. In line with the grounded theory approach which presumes limited preconceived ideas of the research topic, the survey incorporated open-ended questions regarding school closures (e.g. benefits/negative impacts of school closures, concerns and recommendations for face-to-face reopening). This was in order to gather information rich narratives of participants’ perspectives, experiences and recommendations to address the central research question. 

We value your comments regarding the topics discussed in this manuscript and that the broadness of themes captured can be considered heterogenous. The broad and exploratory nature of the open-ended questions, in line with the grounded theory approach, allowed the wide range of impacts and challenges to be captured from those at the frontline of education provision. We have included verbatim quotes in order to illustrate the thematic descriptions. Our findings have been validated by academic research, reports and policy briefs published in the interim of this review period, including at a global level by UNESCO and the WHO. This also includes extensive research within the UK by the National Foundation of Educational Research (NFER) conducted during the COVID-19 pandemic. Findings from the current study are mirrored within a number of reports and policy briefs, with topics such as pupil engagement with learning, impacts on learning and academic outcomes, health and wellbeing, digital barriers, pressures on the education workforce and widening inequalities discussed synchronously (Sharp et al. 2020, Nelson and Sharp 2020, Nelson et al. 2021, Rose et al 2021). We are confident that our grounded theory approach has encapsulated the wide range of impacts reported by school staff and in agreement with the wider literature published since submission. 

Whilst the findings in the current study cover the wide ranging impacts of school closures, we feel this provides a solid foundation for further research to be conducted exploring these topics in greater detail, particularly as we emerge from the period of school closures to the return of face-to-face teaching and learning. 

We have included an additional suggestion of future research to explore these topics in greater detail within the conclusion:

Conclusion. Page: 40, Line 924-927: The broad nature of these recommendations cover the wide ranging impacts of school closures. This calls for future mixed-methods research to explore each topic in greater detail to examine and quantify the depths of these impacts and challenges, particularly since the return of face-to-face education provision.

In relation to your comment regarding the research design and sampling strategy, we feel that estimating the prevalence of issues amongst an entire population of school teachers requiring a large sample would call for a different study that would suit a quantitative design. Sample sizes and prevalence rates are at odds with the premise of qualitative research, which aims to explore the rich lived experiences of participants, as opposed to quantifying prevalence of experiences. In this study, and in line with qualitative research approaches, we were not aiming to be representative/generalisable of the population, and have not claimed this was our approach within the manuscript. Rather, our study aimed to capture and explore the lived experiences of the education workforce in relation to an understudied topic during an emergency period of national school closures. Convenience sampling was most suitable approach during this challenging period for the education workforce and was required in order to conduct rapid qualitative research. As noted above, we have advocated for further research to quantify these impacts. We will continue addressing your comments regarding the sampling strategy in the sub-section below. 

Sharp C, Nelson J,Lucas M, Julius J, McCrone T and Sims D (2020) The challenges facing schools and pupils in September 2020. NFER. Slough. Available at: https://www.nfer.ac.uk/schools-responses-to-covid-19-the-challenges-facing-schools-and-pupils-in-september-2020/

Nelson J and Sharp C (2020) Schools' responses to Covid-19: Key findings from the Wave 1 survey. NFER. Slough. Available at: https://www.nfer.ac.uk/schools-responses-to-covid-19-key-findings-from-the-wave-1-survey/

Nelson J, Lynch S and Sharp C. (2021) Recovery During a Pandemic: the ongoing Impacts of Covid-19 on Schools Serving Deprived Communities. NFER. Slough. Available at: https://www.nfer.ac.uk/recovery-during-a-pandemic-the-ongoing-impacts-of-covid-19-on-schools-serving-deprived-communities/

Rose S, Twist L, Lord P, Rutt S, Badr K, Hope C and Styles B (2021) Impact of school closures and subsequent support strategies on attainment and socio-emotional wellbeing in Key Stage 1: Interim Paper 1. NFER. Slough. Available at: https://www.nfer.ac.uk/impact-of-school-closures-and-subsequent-support-strategies-on-attainment-and-socio-emotional-wellbeing/

3. The empirical foundation

I am in serious doubt to what extent the research design which is based on a convenience sample realized with snowball sampling and similar methods can really be used to “develop a rigorous and robust foundation for shaping and influencing emerging policy and practice in post-Covid education provision” – as argued by the authors. While the identified topics and different perspective identified in the paper are very interesting, no doubt, I wonder whether a different and more systematic sampling approach (either qualitative or quantitative) would have led to very different results. Or - in other words, there is a realistic chance that the teachers who participated in the study might a quite different (special) – and had a very different experience of the Covid-19 pandemic and school closures – than the rest of the teacher population. As a minimum, the authors would need to address this issue in the paper.

The method of convenience sampling was the most suitable approach to recruit participants during a global pandemic, recognising the breadth of lived experiences. We used a number of recruitment methods in order to increase the scope of the survey reach, and we have outlined the three stage approach used for disseminating the survey. The additional layer of snowball sampling aimed to increase the reach of our research study by recruiting school staff through key stakeholders in education, including regional education bodies, who have established relationships with a range of primary schools across Wales. Furthermore, recruitment was facilitated through the HAPPEN Wales primary school network which has existing partnerships with primary school across all local authorities in Wales, serving a wide range of deprivation areas and area characteristics. We are confident that we have accurately and transparently reported our sampling strategy within the manuscript. 

You draw attention to an interesting point regarding the potential of a different sampling strategy yielding different results. It is indeed possible that those schools engaged with HAPPEN, regional education groups and those that participated in the survey could be considered as ‘research engaged’. This remains a challenge in all school-based research, and particularly during a time in which the education workforce was experiencing significantly higher working demands and increased stress. Perhaps most usefully, many of these schools and staff that participated in the study are also likely to be engaged with emerging policy design, for example the new Curriculum for Wales due to be rolled out in 2022, which has received significant input from ‘pioneer schools’ across Wales to its design and development. We aim to further extend collaborations through the HAPPEN network for future research projects to engage with wider groups and capture perspectives from those harder to reach.

Whilst we acknowledge your comment regarding different sampling strategies, we are still confident that the convenience and snowball strategy employed in our study is reflective of the perspectives of key education practitioners. Furthermore, our findings have since been validated in wider research across the UK since original submission (Sharp et al. 2020, Nelson and Sharp 2020, Nelson et al. 2021, Rose et al. 2021). Qualitative research allows an exploration of people’s experiences of education systems and policies including school closures and policy regarding the COVID response. This helps us to understand how these responses/policies can be shaped. We believe our actionable findings contribute to this understanding and have captured the perspectives of the education workforce in agreement with other literature. 

To address your comments regarding sampling, we have included additional discussions in the limitations section acknowledging the challenges of convenience sampling in reporting findings, and that those that did not participate may have experienced the period of school closures differently and thus, reported different perspectives. We feel this is an important point to include and we are grateful that you have drawn this to our attention. 

Strengths and limitations. Page 39, line 885-890: The perspectives captured in this study may not account for the full breadth of lived experiences of all primary school staff during school closures and reopening. It is possible that those that participated in this study work within schools that are ‘research engaged’. Those that did not participate may have different experiences and perspectives during this period to those reported in this study.

Best wishes,

Dr Emily Marchant

---

## [Decision Letter · Decision Letter 3]

10 Nov 2021

Primary school staff perspectives of school closures due to COVID-19, experiences of schools reopening and recommendations for the future: a qualitative survey in Wales.

PONE-D-20-34984R3

Dear Dr. Marchant,

We’re pleased to inform you that your manuscript has been judged scientifically suitable for publication and will be formally accepted for publication once it meets all outstanding technical requirements.

Kind regards,

Amanda A. Webster

Academic Editor

PLOS ONE

Additional Editor Comments (optional):

Reviewers' comments:

Reviewer's Responses to Questions

**Comments to the Author**

1. If the authors have adequately addressed your comments raised in a previous round of review and you feel that this manuscript is now acceptable for publication, you may indicate that here to bypass the “Comments to the Author” section, enter your conflict of interest statement in the “Confidential to Editor” section, and submit your "Accept" recommendation.

Reviewer #1: All comments have been addressed

2. Is the manuscript technically sound, and do the data support the conclusions?

Reviewer #1: Yes

3. Has the statistical analysis been performed appropriately and rigorously? 

Reviewer #1: Yes

4. Have the authors made all data underlying the findings in their manuscript fully available?

Reviewer #1: Yes

5. Is the manuscript presented in an intelligible fashion and written in standard English?

Reviewer #1: Yes

6. Review Comments to the Author

Reviewer #1: Well done on finishing your paper and addressing the reviewers comments. My only comment is in regard to the theoretical underpinnings ---you have a range of ideas here --grounded theory, pragmatism and interpretivism-----my suggestion is for you to pick either interpretivism or pragmatism, but not both --as they are different concepts.

All the best with this valuable study.

Lynn

7. PLOS authors have the option to publish the peer review history of their article (what does this mean?). If published, this will include your full peer review and any attached files.

Reviewer #1: **Yes: **Dr Lynn Sheridan

---

## [Editor Report · Acceptance letter]

22 Nov 2021

PONE-D-20-34984R3 

Primary school staff perspectives of school closures due to COVID-19, experiences of schools reopening and recommendations for the future: a qualitative survey in Wales 

Dear Dr. Marchant:

I'm pleased to inform you that your manuscript has been deemed suitable for publication in PLOS ONE. Congratulations! Your manuscript is now with our production department. 

Kind regards, 

on behalf of

Dr. Amanda A. Webster 

Academic Editor

PLOS ONE